# SWI/SNF senses carbon starvation with a pH-sensitive low-complexity sequence

**J Ignacio Gutierrez[1†], Gregory P Brittingham[2], Yonca Karadeniz[3], Kathleen D Tran[4], Arnob Dutta[4], Alex S Holehouse[5,6], Craig L Peterson[3], Liam J Holt[2]***

[1]Department of Molecular and Cell Biology, University of California, Berkeley, Berkeley, United States; [2]Institute for Systems Genetics, New York University Grossman School of Medicine, New York, United States; [3]Program in Molecular Medicine, University of Massachusetts Medical School, Worcester, United States; [4]Department of Cell and Molecular Biology, University of Rhode Island, Kingston, United States; [5]Center for Science and Engineering of Living Systems (CSELS), Washington University in St. Louis, St Louis, United States; [6]Department of Biochemistry and Molecular Biophysics, Washington University School of Medicine, St. Louis, United States

**\*For correspondence:**
Liam.Holt@nyulangone.org

**Present address:** [†]Weill Cornell Medicine, New York, United States

**Abstract** It is increasingly appreciated that intracellular pH changes are important biological signals. This motivates the elucidation of molecular mechanisms of pH sensing. We determined that a nucleocytoplasmic pH oscillation was required for the transcriptional response to carbon starvation in *Saccharomyces cerevisiae*. The SWI/SNF chromatin remodeling complex is a key mediator of this transcriptional response. A glutamine-rich low-complexity domain (QLC) in the *SNF5* subunit of this complex, and histidines within this sequence, was required for efficient transcriptional reprogramming. Furthermore, the *SNF5* QLC mediated pH-dependent recruitment of *SWI/SNF* to an acidic transcription factor in a reconstituted nucleosome remodeling assay. Simulations showed that protonation of histidines within the *SNF5* QLC leads to conformational expansion, providing a potential biophysical mechanism for regulation of these interactions. Together, our results indicate that pH changes are a second messenger for transcriptional reprogramming during carbon starvation and that the *SNF5* QLC acts as a pH sensor.

## Editor's evaluation

This study has considerable merit in providing evidence that the Q-rich low-complexity domain in Snf5, and the histidine residues located therein, functions as a sensor of the drop in intracellular pH that accompanies glucose starvation to mediate SWI/SNF recruitment and transcriptional activation of the battery of genes derepressed under these conditions in order to reprogram carbon utilization. The work is multifaceted in combining yeast genetics, single-cell assays of gene expression and intracellular pH, genome-wide analysis of gene expression changes by RNA-seq, and in vitro biophysical analysis of activator-dependent SWI/SNF recruitment and nucleosome remodeling in a purified system.

## Introduction

Biological processes are inherently sensitive to the solution environment in which they occur. A key regulated parameter is intracellular pH (pH$_i$), which influences all biological processes by determining the protonation state of titratable chemical groups. These titratable groups are found across many

biological molecules, from small-molecule osmolytes to the side chains of amino acids. While early work suggested that $pH_i$ was a tightly constrained cellular parameter, more recent technologies have revealed that $pH_i$ can vary substantially in both space and time (**Llopis et al., 1998**; **Seksek and Bolard, 1996**). Moreover, changes in $pH_i$ can regulate metabolism (**Busa and Nuccitelli, 1984**; **Young et al., 2010**), development (**Needham and Needham, 1997**), proliferation (**Busa and Crowe, 1983**), and cell fate (**Okamoto, 1994**), among other processes. Intriguingly, stress-associated intracellular acidification appears to be broadly conserved, suggesting that a drop in $pH_i$ is a primordial mechanism to coordinate the general cellular stress response (**Drummond et al., 1986**; **Gores et al., 1989**; **Munder et al., 2016**; **O'Sullivan and Condon, 1997**; **Triandafillou et al., 2020**; **Yao and Haddad, 2004**).

The budding yeast *Saccharomyces cerevisiae* is adapted to an acidic external environment ($pH_e$), and optimal growth media is typically at pH 4.0–5.5. The plasma membrane (Pma1) and vacuolar (Vma1) ATPases maintain near neutral $pH_i$ of ~7.8 by pumping protons out of the cell and into the vacuole, respectively (**Martínez-Muñoz and Kane, 2008**). When cells are starved for carbon, these pumps are inactivated, leading to a rapid acidification of the intracellular space to pH ~6 (**Kane, 1995**; **Orij et al., 2009**). This decrease in intracellular $pH_i$ is crucial for viability upon carbon starvation and is thought to conserve energy, leading to storage of metabolic enzymes in filamentous assemblies (**Petrovska et al., 2014**), reduction of macromolecular diffusion (**Joyner et al., 2016**; **Munder et al., 2016**), decreased membrane biogenesis (**Young et al., 2010**), and possibly the noncovalent cross-linking of the cytoplasm into a solid-like material state (**Joyner et al., 2016**; **Munder et al., 2016**). These studies suggest that many physiological processes are inactivated when $pH_i$ drops. However, some processes must also be upregulated during carbon starvation to enable adaptation to this stress. These genes are referred to as 'glucose-repressed genes' as they are transcriptionally repressed in the presence of glucose (**DeRisi et al., 1997**; **Zid and O'Shea, 2014**). Recently, evidence was presented of a positive role for acidic $pH_i$ in stress-gene induction: transient acidification is required for induction of the transcriptional heat-shock response in some conditions (**Triandafillou et al., 2020**). However, the molecular mechanisms by which the transcriptional machinery senses and responds to pH changes remain mysterious.

The *Sucrose Non-Fermenting* genes (*SNF*) were among the first genes found to be required for induction of glucose-repressed genes (**Neigeborn and Carlson, 1984**). Several of these genes were later identified as members of the SWI/SNF complex (**Abrams et al., 1986**; **Carlson, 1987**), an 11-subunit chromatin remodeling complex that is highly conserved from yeast to mammals (**Chiba et al., 1994**; **Peterson et al., 1994**; **Peterson and Herskowitz, 1992**). The SWI/SNF complex affects the expression of ~10% of the genes in *S. cerevisiae* during vegetative growth (**Sudarsanam et al., 2000**). Upon carbon starvation, most genes are downregulated, but a set of glucose-repressed genes, required for utilization of alternative energy sources, are strongly induced (**Zid and O'Shea, 2014**). The SWI/SNF complex is required for the efficient expression of several hundred stress-response and glucose-repressed genes, implying a possible function in pH-associated gene expression (**Biddick et al., 2008a**; **Sudarsanam et al., 2000**). However, we still lack evidence for a direct role for SWI/SNF components in the coordination of pH-dependent transcriptional programs or a mechanism through which pH sensing may be achieved.

10/11 subunits of the SWI/SNF complex contain large intrinsically disordered regions (**Figure 1— figure supplement 1**), and in particular, 4/11 SWI/SNF subunits contain glutamine-rich low-complexity sequences (QLCs). QLCs are present in glutamine-rich transactivation domains (**Kadonaga et al., 1988**; **Kadonaga et al., 1987**) some of which, including those found within SWI/SNF, may bind to transcription factors (**Prochasson et al., 2003**) or recruit transcriptional machinery (**Geng et al., 2001**; **Janody et al., 2001**; **Laurent et al., 1990**). Intrinsically disordered regions lack a fixed three-dimensional structure and can be highly responsive to their solution environment (**Holehouse and Sukenik, 2020**; **Moses et al., 2020**). Moreover, the SWI/SNF QLCs contain multiple histidine residues. Given that the intrinsic $pK_a$ of the histidine sidechain is 6.9 (**Whitten et al., 2005**), we hypothesized that these glutamine-rich low-complexity regions might function as pH sensors in response to variations in $pH_i$.

In this study, we elucidate *SNF5* as a pH-sensing regulatory subunit of SWI/SNF. *SNF5* is over 50% disordered and contains the largest QLC of the SWI/SNF complex. This region is 42% glutamine and contains seven histidine residues. We investigated the relationship between the *SNF5* QLC and the cytosolic acidification that occurs during acute carbon starvation. By single-cell analysis, we found that

intracellular pH (pH$_i$) is highly dynamic and varies between subpopulations of cells within the same culture. After an initial decrease to pH$_i$ ~ 6.5, a subset of cells recovered their pH$_i$ to ~7. This transient acidification followed by recovery was required for expression of glucose-repressed genes. The *SNF5* QLC and four embedded histidines were required for rapid gene induction. SWI/SNF complex histone remodeling activity was robust to pH changes, but recruitment of the complex to a model transcription factor was pH-sensitive, and this recruitment was mediated by the *SNF5* QLC and histidines within. All-atom simulations indicated that histidine protonation causes a conformational expansion of the *SNF5* QLC, perhaps enabling interaction with a different set of transcription factors and driving recruitment to the promoters of glucose-repressed genes. Thus, we propose changes in histidine charge within QLCs as a mechanism to sense pH changes and instruct transcriptional reprograming during carbon starvation.

## Results

### Induction of *ADH2* upon glucose starvation requires the *SNF5* glutamine-rich low-complexity sequence with native histidines

The SWI/SNF chromatin remodeling complex subunit *SNF5* has a large low-complexity region at its N-terminus that is enriched for glutamine, the sequence of which is shown in *Figure 1A*. This sequence contains seven histidine residues, and we noticed a frequent co-occurrence of histidines within and adjacent to glutamine-rich low-complexity sequences (QLCs) of many proteins. Inspection of the sequence properties of proteins, especially through the lens of evolution, can provide hints as to functionally important features. Therefore, we analyzed the sequence properties of all glutamine-rich low-complexity sequences (QLCs) in the proteomes of several species.

We defined QLCs as protein subsequences with a minimum of 25% glutamine residues, a maximum interruption between any two glutamine residues of 17 residues, and a minimum overall length of 15 residues. These parameters were optimized empirically based on the features of glutamine-rich regions in the *S. cerevisiae proteme* (see Materials and methods and *Figure 1—figure supplement 2*). By these criteria, the S288c *S. cerevisiae* strain had 144 QLCs (*Supplementary file 1*). We found that proline and histidine were enriched (>50–100%-fold higher than average proteome abundance) in yeast QLCs (*Figure 1B*), with similar patterns found in *Dictyostelium discoideum*, and *Drosophila melanogaster* proteomes (*Figure 1—figure supplement 3*). Enrichment for histidine within QLCs was previously described across many *Eukaryotes* using a slightly different method (*Ramazzotti et al., 2012*). Interestingly, the codons for glutamine are a single base pair mutation away from proline and histidine. However, they are similarly adjacent to lysine, arginine, glutamate, and leucine, yet QLCs are depleted for lysine, arginine, and glutamate, suggesting that the structure of the genetic code is insufficient to explain the observed patterns of amino acids within QLCs. We also considered the possibility that histidines might be generally enriched in low-complexity sequences. In fact, this is not the case: histidines are 50% more abundant in yeast QLCs than in all other low-complexity sequences identified using Wootton–Federhen complexity (see Materials and methods). Thus, histidines are a salient feature of QLCs.

The N-terminus of *SNF5* contains one of the largest QLCs in the yeast proteome and is in the top 3 QLCs in terms of number of histidines (*Figure 1—figure supplement 2E and F*). We compared the sequences of Snf5 N-terminal domains taken from 20 orthologous proteins from a range of *Ascomycota* (a fungal phylum) (*Figure 1—figure supplement 4*, *Supplementary file 2*). Despite the relatively poor sequence conservation across the N-terminal disordered regions in *SNF5* (*Figure 1—figure supplement 4A*), every region consisted of at least 18% glutamine (max 43%) and all possessed multiple histidine residues (*Figure 1—figure supplement 4B*, *Supplementary file 2*; the phylogeny considered and the total number of QLCs for each species are shown in *Figure 1—figure supplement 4C*). A broader survey of the tree of life (*Figure 1—figure supplement 5*) indicates that the *SNF5* QLC was likely gained in the lineage leading to the *Ascomycota* and is not present in most *Metazoa* (animals). In summary, enrichment for glutamine residues interspersed with histidine residues appears to be a conserved sequence feature, both in QLCs, in general, and in the N-terminus of *SNF5,* in particular, implying a possible functional role (*Zarin et al., 2019*).

To further investigate the functional importance of the glutamine-rich N-terminal domain in *SNF5,* we engineered three *SNF5* mutant strains: a complete deletion of the *SNF5* gene (*snf5Δ*); a deletion

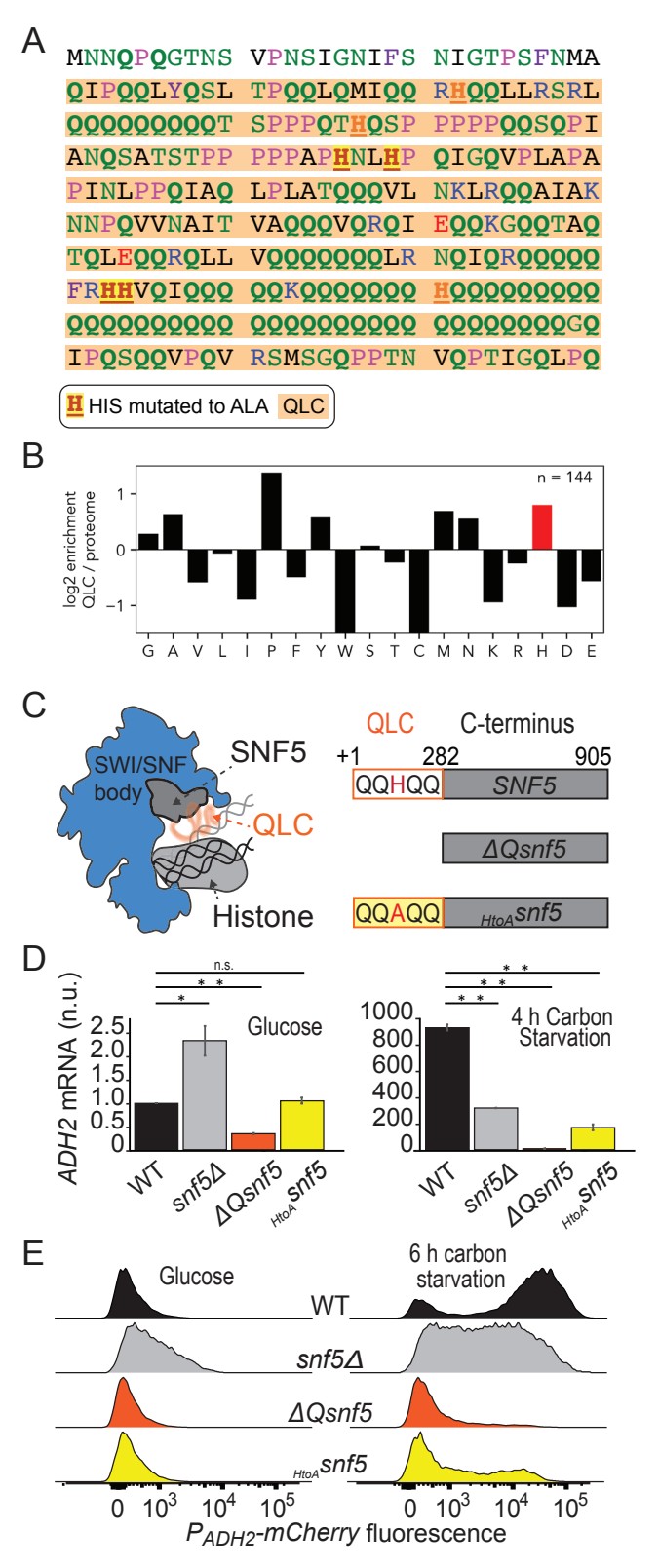

**Figure 1.** Efficient induction of *ADH2* upon glucose starvation requires the *SNF5* glutamine-rich low-complexity sequence with native histidines. (**A**) Sequence of the N-terminal low-complexity domain of *SNF5*. This domain was deleted in the *ΔQsnf5* strain. The glutamine-rich domain is highlighted in orange. The 4/7 histidines that were mutated to alanine in the *HtoASNF5* allele are highlighted in red. (**B**) The log$_2$ of the frequency of each amino

*Figure 1 continued on next page*

*Figure 1 continued*

acid within QLCs divided by the global frequency of each amino acid in the proteome (*S. cerevisiae*). Values > 0 indicate enrichment in QLCs. (**C**) Left: schematic of the SWI/SNF complex engaged with a nucleosome. The *SNF5* C-terminus is shown in gray, while the disordered N-terminal QLC is shown in orange. Right: schematic of the three main *SNF5* alleles used in this study. (**D**) RT-qPCR results assessing levels of endogenous *ADH2* mRNA in four strains grown in glucose (left) or after 4 hr of glucose starvation (right). Note: y-axes are different for each plot. (**E**) Representative histograms (10,000 cells) showing the fluorescent signal from a $P_{ADH2}$-*mCherry* reporter gene for four strains grown in glucose (left) or after 6 hr of glucose starvation (right). Statistical tests are Bonferroni-corrected *t*-tests, *p<0.05, **p<0.01, n.s., not significant.

The online version of this article includes the following source data and figure supplement(s) for figure 1:

**Figure supplement 1.** The SWI/SNF complex has 10/11 subunits with significant disorder.

**Figure supplement 2.** Identification and analysis of glutamine-rich low-complexity sequences (QLCs).

**Figure supplement 3.** Histidines are enriched in glutamine-rich low-complexity sequences.

**Figure supplement 4.** The *SNF5* N-terminal glutamine-rich low-complexity domain (with embedded histidines) is broadly conserved across *Ascomycota*.

**Figure supplement 5.** The *SNF5* N-terminal glutamine-rich low-complexity domain was probably gained in the fungal lineage.

**Figure supplement 6.** The *SNF5* QLC is important for recovery from carbon starvation.

**Figure supplement 7.** Mutation of the *SNF5* QLC does not lead to protein degradation or loss of SWI/SNF complex integrity.

**Figure supplement 7—source data 1.** The entire SWI/SNF complex copurifies with SNF2 in all strains and conditions.

**Figure supplement 8.** Efficient recruitment of the SWI/SNF complex to the *ADH2* promoter depends upon pH, the *SNF5* QLC and histidines within.

**Figure supplement 9.** The *SNF5* QLC and embedded histidines are required for efficient *ADH2* induction upon carbon starvation.

---

of the N-terminal QLC (*ΔQsnf5*); and an allele with four histidines within the QLC mutated to alanine (*HtoAsnf5*) (**Figure 1A and C**).

As previously reported (**Laurent et al., 1990**), *snf5Δ* strains grew slowly (**Figure 1—figure supplement 6A**). In contrast, growth rates of *ΔQsnf5* and *HtoAsnf5* were similar to WT during continuous growth in either fermentable (glucose) or poor (galactose or galactose/ethanol) carbon sources (**Figure 1—figure supplement 6A–D**) and showed minimal defects when grown in glucose, carbon-starved for 24 hr, and then reinoculated into glucose media. However, a strong growth defect was revealed for *ΔQsnf5* and *HtoAsnf5* strains when cells were carbon-starved for 24 hr and then switched to a poor carbon source (**Figure 1—figure supplement 6E and F**), suggesting that the *SNF5* QLC is important for adaptation to new carbon sources. Deletion of the *SNF5* gene has been shown to disrupt the architecture of the SWI/SNF complex, leading to loss of other subunits (**Peterson et al., 1994**; **Yang et al., 2007**). To test if deletion of the QLC leads to loss of *Snf5p* protein or failure to incorporate into SWI/SNF, we immunoprecipitated the SWI/SNF complex from strains with a tandem affinity purification (TAP) tag at the C-terminal of the core *SNF2* subunit. We found that the entire SWI/SNF complex remained intact in both the *ΔQsnf5* and *HtoAsnf5* strains (**Figure 1—figure supplement 7A**). Silver stains of the untagged *Snf5p* and Western blotting of TAP-tagged *SNF5* (**Puig et al., 2001**) strains showed that all *SNF5* alleles were expressed at similar levels to wild-type both in glucose and upon carbon starvation (**Figure 1—figure supplement 7B**). Together, these results show that deletion of the *SNF5* QLC is distinct from total loss of the *SNF5 gene* and that this N-terminal sequence is important for efficient recovery from carbon starvation.

We hypothesized that slow recovery of *ΔQsnf5* and *HtoAsnf5* strains after carbon starvation was due to a failure in transcriptional reprogramming. The alcohol dehydrogenase *ADH2* gene is normally repressed in the presence of glucose and strongly induced upon carbon starvation. This regulation depends on SWI/SNF activity (**Peterson and Herskowitz, 1992**). Therefore, we used *ADH2* as a model gene to test our hypothesis. We assayed SWI/SNF occupancy at the *ADH2* promoter by chromatin immunoprecipitation (ChIP) of SWI/SNF complexes with a TAP-tag on the C-terminus of the *SNF2* subunit from strains with various *SNF5* alleles, followed by quantitative PCR (qPCR). These

experiments showed that the wild-type complex is robustly recruited to the *ADH2* promoter upon carbon starvation (*Figure 1—figure supplement 8*). However, this recruitment is reduced in *ΔQsnf5* and *HtoAsnf5* strains.

Next, we assayed transcription of the *ADH2* gene using reverse transcriptase quantitative polymerase chain reaction (RT-qPCR). We found that robust *ADH2* expression after acute carbon starvation was dependent on the *SNF5* QLC and the histidines within (*Figure 1D*). This defect was far stronger in the *ΔQsnf5* and *HtoAsnf5* strains than in *snf5Δ* strains; *snf5Δ* strains did not completely repress *ADH2* expression in glucose and showed partial induction upon carbon starvation, while *ΔQsnf5* strains tightly repressed *ADH2* in glucose (similar to WT), but completely failed to induce expression upon starvation (*Figure 1D*). These results suggest a dual role for *SNF5* in *ADH2* regulation, both contributing to strong repression in glucose and robust induction upon carbon starvation. The *ΔQsnf5* and *HtoAsnf5* alleles separate these functions, maintaining WT-like repression while showing a strong defect in induction.

The RT-qPCR and ChIP assays report on the average behavior of a population. To enable single-cell analysis, we engineered a reporter strain with the mCherry (*Shaner et al., 2004*) fluorescent protein under the control of the *ADH2* promoter integrated into the genome immediately upstream of the endogenous *ADH2* locus (*Figure 1E*, *Figure 1—figure supplement 9A*). We found high cell-to-cell variation in the expression of this reporter in WT strains: after 6 hr of glucose starvation, $P_{ADH2}$-*mCherry* expression was bimodal; about half of the cells had high mCherry fluorescence and half were low. This bimodality was strongly dependent on preculture conditions and was most apparent upon acute withdrawal of carbon from early log-phase cells that had grown for >16 hr with optical density at 600 nm (O.D.) never exceeding 0.3 (see Materials and methods). If cells became partly saturated at any time during preculture, *ADH2* induction was more rapid and uniform. Complete deletion of *SNF5* eliminated this bimodal expression pattern; again, low levels of expression were apparent in glucose and induction during starvation was attenuated. As in the RT-qPCR analysis, the *ΔQsnf5* strain completely failed to induce the $P_{ADH2}$-*mCherry* reporter at this time point and mutation of four central histidines to alanine was sufficient to mostly abrogate expression (*Figure 1E*). Mutation of a further two histidines had little additional effect (*Figure 1—figure supplement 9B–D*). Taken together, these results suggest that the dual function of *SNF5* leads to switch-like control of *ADH2* expression. In glucose, *SNF5* helps repress *ADH2*. Upon carbon starvation, *SNF5* is required for efficient induction of *ADH2*. The *SNF5* QLC and histidine residues within seem to be crucial for switching between these states.

## The *SNF5* QLC is required for *ADH2* expression and recovery of neutral pH

Multiple stresses, including glucose starvation, have been shown to cause a decrease in the pH of the cytoplasm and nucleus (nucleocytoplasm) (*Dechant et al., 2014*; *Gores et al., 1989*; *Triandafillou et al., 2020*; *Yao and Haddad, 2004*). Here, we refer to nucleocytoplasmic pH as *intracellular pH* ($pH_i$). To investigate the relationship between *ADH2* expression and $pH_i$, and how these factors depend upon *SNF5*, we engineered strains bearing both the ratiometric fluorescent pH reporter, pHluorin (*Miesenböck et al., 1998*), and the $P_{ADH2}$-*mCherry* reporter. To calibrate the pHluorin sensor, we calculated the ratio of intensities of fluorescence emission after excitation with 405 and 488 nm light in cells that were ATP-depleted and permeabilized in media of known pH. We obtained a near linear relationship between ratios of fluorescence intensity and pH (*Figure 2—figure supplement 1*, Materials and methods). Therefore, these strains allowed us to simultaneously monitor $pH_i$ and expression of *ADH2*.

Wild-type cells growing exponentially in 2% glucose had a $pH_i$ of ~7.8. Upon acute carbon starvation, cells rapidly acidified to $pH_i$ ~ 6.5. Then, during the first hour, two populations arose: an acidic population ($pH_i$ ~ 5.5), and a second population that recovered to $pH_i$ ~ 7 (*Figure 2A*). Cells at $pH_i$ 7 proceeded to strongly induce expression of the $P_{ADH2}$-*mCherry* reporter, while cells at $pH_i$ 5.5 did not. We used fluorescence-activated cell sorting (FACS) to separate these two populations and found that cells that neither recovered neutral pH nor expressed the $P_{ADH2}$-*mCherry* reporter had lower fitness relative to the $P_{ADH2}$-*mCherry*-inducing population, as indicated by lower rates of proliferation on both rich and poor carbon sources, and lower tolerance of heat stress (*Figure 2—figure supplement 2*). After 8 hr of glucose starvation, >70% of wild-type cells had induced *ADH2* (*Figure 2A and C*).

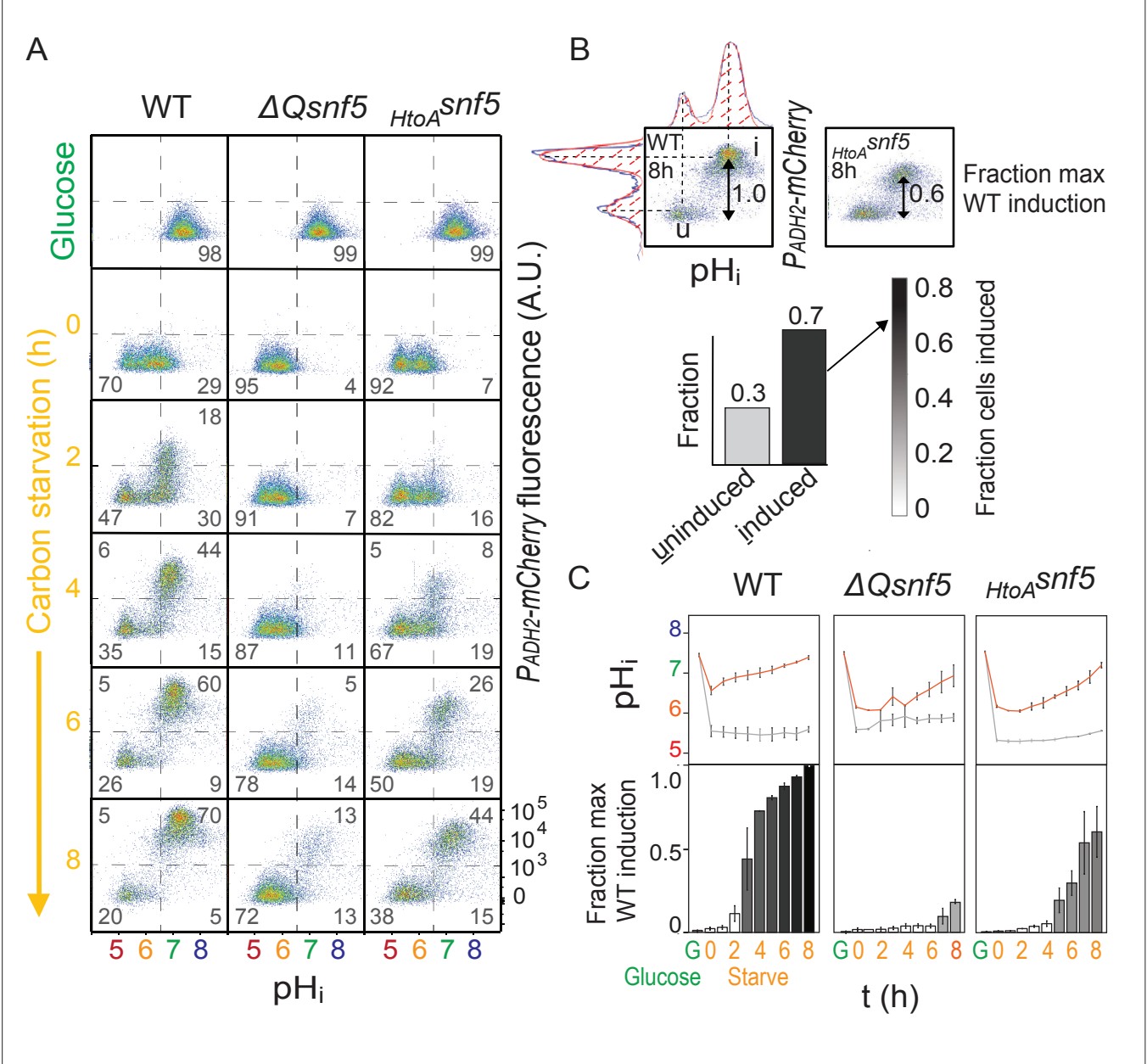

**Figure 2.** The *SNF5* QLC is required for *ADH2* expression and recovery of neutral pH. (**A**) Representative flow cytometry for WT, *ΔQsnf5*, or *HtoAsnf5* strains: the x-axis shows nucleocytoplasmic pH (pH$_i$), while the y-axis shows fluorescence from the $P_{ADH2}$-*mCherry* reporter. Panels show cells grown in glucose (top) and then (second to bottom) after 0–8 hr of acute glucose starvation. Percentage of cells in each quadrant is indicated by gray numbers. (**B**) Schematic of quantification scheme: raw data from (**A**) was fit to a single or double Gaussian curve determined by a least-residuals method. (**C**) Quantification of pH$_i$ and $P_{ADH2}$-*mCherry* expression during acute starvation. The median of each Gaussian for pHi is plotted in (**C**, top), black and gray lines are from induced and uninduced populations, respectively. The height of bars in (**C**, bottom) indicates the fraction of maximal $P_{ADH2}$-*mCherry* reporter gene expression (WT cells, 8 hr glucose starvation) The darkness of the bars indicates the fraction of the population in the induced versus uninduced state. Mean and standard deviation of three biological replicates are shown.

The online version of this article includes the following figure supplement(s) for figure 2:

**Figure supplement 1.** Examples of calibration curves to measure cytosolic pH using pHluorin.

**Figure supplement 2.** Cells that fail to induce $P_{ADH2}$-*mCherry* had lower fitness relative to the inducing population.

**Figure supplement 3.** All strains ultimately express some amount of $P_{ADH2}$-*mCherry* reporter.

**Figure supplement 4.** *snf5Δ* strains only had a slight delay in expression of the $P_{ADH2}$-*mCherry* reporter.

**Figure supplement 5.** Recovery of pH$_i$ requires new protein translation.

We next analyzed cells harboring mutant alleles of the QLC of *SNF5*. Similarly to WT, both *ΔQsnf5* and $_{HtoA}snf5$ strains rapidly acidified upon carbon starvation. However, these strains were defective in subsequent neutralization of pH$_i$ and in the expression of $P_{ADH2}$-*mCherry*. At the 4 hr time point, >95% of both *ΔQsnf5* and $_{HtoA}snf5$ cells remained acidic with no detectable expression, while >60% of wild-type cells had neutralized and expressed mCherry (*Figure 2A and C*). Eventually, after 24 hr, the majority of mutant cells neutralized to pH$_i$ ~7 and induced expression of $P_{ADH2}$-*mCherry* (*Figure 2— figure supplement 3*). Again, complete deletion of *SNF5* led to less severe phenotypes than the *ΔQsnf5* and $_{HtoA}snf5$ alleles with only a modest delay in $P_{ADH2}$-*mCherry* expression (*Figure 2—figure supplement 4*), suggesting that *SNF5* plays both activating and inhibitory roles in *ADH2* expression. Thus, the *SNF5* QLC and histidines within are required for the rapid dynamics of both transient acidification and transcriptional induction of $P_{ADH2}$-*mCherry* upon acute carbon starvation.

We hypothesized that mutant cells might fail to recover from acidification because transcripts controlled by SWI/SNF are responsible for pH$_i$ recovery. In this model, SWI/SNF drives expression of a set of genes that must be both transcribed and translated. To test this idea, we measured pH$_i$ in WT cells during carbon starvation in the presence of the cyclohexamine to prevent translation of new transcripts. In these conditions, we found that cells experienced a drop in pH$_i$ but were unable to recover neutral pH (*Figure 2—figure supplement 5*). Thus, new gene expression is required for recovery of pH$_i$.

## Transient acidification is required for *ADH2* induction upon carbon starvation

The acidification of the yeast nucleocytoplasm has been shown to depend upon an acidic extracellular pH (pH$_e$). We took advantage of this fact to manipulate the changes in pH$_i$ that occur upon carbon starvation. Cell viability was strongly dependent on pH$_e$, decreasing drastically when cells were starved for glucose in media at pH ≥ 7.0 for 24 hr (*Figure 3—figure supplement 1*). Expression of $P_{ADH2}$-*mCherry* expression was also highly dependent on pH$_e$, especially in *SNF5* QLC mutants (*Figure 3A*, *Figure 3—figure supplement 2*). WT cells failed to induce $P_{ADH2}$-*mCherry* at pH$_e$ ≥ 7, but induced strongly at pH$_e$ ≤ 6.5. RT-qPCR showed similar behavior for the endogenous *ADH2* transcript (*Figure 3—figure supplement 3*). ChIP experiments indicated that recruitment of SWI/SNF to the *ADH2* promoter was also reduced when starvation was performed with media buffered to pH$_e$ 7.5 (*Figure 1—figure supplement 7*). Furthermore, we found that the nucleocytoplasm of all strains failed to acidify when the environment was held at pH$_e$ ≥ 7 (*Figure 3—figure supplement 4*). Therefore, we conclude that an acidic extracellular environment is required for a decrease in intracellular pH upon carbon starvation, and that this intracellular acidification is required for activation of *ADH2* transcription.

Given that intracellular acidification is necessary for *ADH2* promoter induction, we next wondered if it was sufficient. First, we used the membrane-permeable sorbic acid to allow intracellular acidification but prevent pH$_i$ recovery. These cells failed to induce $P_{ADH2}$-*mCherry*, indicating that nucleocytoplasmic acidification is not sufficient; subsequent neutralization is also required. Carbon starvation at pH$_e$ 7.4 prevented transient acidification and likewise prevented expression (*Figure 3B*, *Figure 3— figure supplement 3*). Cells that were first held at pH$_e$ 7.4, preventing initial acidification, and then switched to pH$_e$ 5, thereby causing late acidification, failed to express mCherry after 6 hr. Finally, starvation at pH$_e$ 5 for 2 hr followed by a switch to pH$_e$ 7.4, with a corresponding increase in pH$_i$, led to robust $P_{ADH2}$-*mCherry* expression. Together, these results suggest that transient acidification immediately upon switching to carbon starvation followed by recovery to neutral pH$_i$ is the signal for the efficient induction of $P_{ADH2}$-*mCherry*.

Deletion of the *SNF5* QLC leads to both failure to neutralize pH$_i$ and loss of *ADH2* expression. We therefore wondered if forcing cells to neutralize pH$_i$ would rescue *ADH2* expression in a *ΔQsnf5* strain. This was not the case: the *ΔQsnf5* strain still fails to express $P_{ADH2}$-*mCherry*, even if we recapitulate normal intracellular transient acidification (*Figure 3B*, right). Therefore, the *SNF5* QLC is required for normal kinetics of transient acidification *and* for additional steps in *ADH2* gene activation.

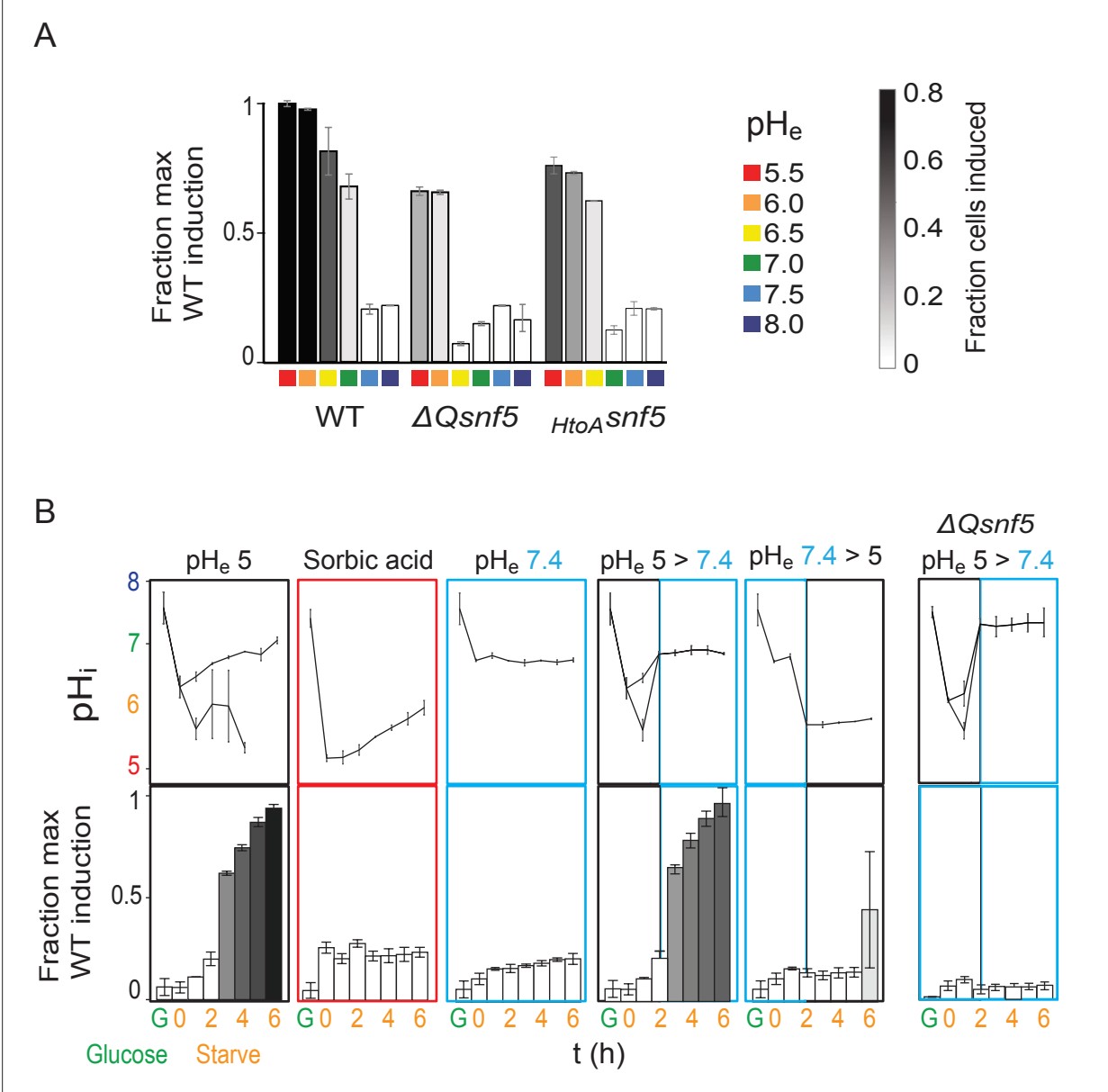

**Figure 3.** Transient acidification is required for *ADH2* induction upon carbon starvation. (**A**) Expression of $P_{ADH2}$-*mCherry* reporter gene in WT, *ΔQsnf5*, or *HtoAsnf5* strains 8 hr after acute carbon starvation in media titrated to various pH (pH$_e$, see legend, right). Bar height indicates the fraction of maximal $P_{ADH2}$-*mCherry* reporter gene expression (WT cells, pH$_e$ 5.5). The darkness of the bars indicates the fraction of the population in the induced versus uninduced state (see legend, right). (**B**) Time courses of glucose starvation with media manipulations to perturb the intracellular pH response, either by changing media pH (pH$_e$) or by adding sorbic acid. Top panels show nucleocytoplasmic pH (pH$_i$), black and gray lines from induced and uninduced populations, respectively. Bottom panels quantify expression of the $P_{ADH2}$-*mCherry* reporter gene (as in **A**). All strains are WT except for the far-right panels, which are from a *ΔQsnf5* strain.

The online version of this article includes the following figure supplement(s) for figure 3:

**Figure supplement 1.** Deletion of the N-terminal glutamine-rich domain of *SNF5* renders cells hypersensitive to starvation at suboptimal extracellular pH.

**Figure supplement 2.** $P_{ADH2}$-*mCherry* induction requires an acidic extracellular environment and the *SNF5* QLC.

**Figure supplement 3.** Expression of the endogenous *ADH2* mRNA requires an acidic extracellular environment and the *SNF5* QLC.

**Figure supplement 4.** Transient acidification of cells requires an acidic extracellular environment.

## The *SNF5* QLC and acidification of the nucleocytoplasm are required for efficient widespread transcriptional reprogramming upon carbon starvation

We wondered if transient acidification and the QLC of *SNF5* were important for transcriptional reprogramming on a genome-wide scale. To test this, we performed Illumina RNA-sequencing analysis on triplicates of each strain (WT, *ΔQsnf5*, *HtoAsnf5*) either growing exponentially in glucose or after acute carbon starvation for 4 hr at pH$_e$ 5. In addition, to test the pH dependence of the transcriptional response, we analyzed WT strains carbon-starved at pH$_e$ 7, which prevents intracellular acidification (*Figure 3B*, *Figure 3—figure supplement 4*).

Principal component analysis showed tight clustering of all exponentially growing samples, indicating that mutation of the QLC of *SNF5* does not strongly affect gene expression in rich media (*Figure 4A*). In contrast, there are greater differences between wild-type strains with mutant *SNF5* alleles upon glucose starvation. The genes that accounted for most variation (the first two principal components) were involved in carbon transport, metabolism, and stress responses. We defined a set of 89 genes that were induced (greater than threefold) and 60 genes that were downregulated (greater than threefold) in WT strains upon starvation in media titrated to pH$_e$ 5. Many of these genes were poorly induced in *ΔQsnf5* and *HtoAsnf5* mutants, as well as in WT strains starved in media titrated to suboptimal pH$_e$ 7 (*Figure 4B*). *Figure 4C and D* show transcriptional differences between glucose-starved strains as volcano plots, emphasizing large-scale differences between WT and *ΔQsnf5* strains, and similarities between *ΔQsnf5* and *HtoAsnf5*.

We next performed hierarchical clustering analysis (Euclidean distance) of the 149 genes that are strongly differentially expressed between strains or at suboptimal pH$_e$ 7 (*Figure 4E*). Based on this clustering and some manual curation, we assigned these genes to four groups. Group 1 genes (n = 42) were activated in starvation in an *SNF5* QLC and pH-dependent manner. They are strongly induced in WT, but induction is attenuated both in mutants of the *SNF5* QLC and when the transient acidification of pH$_i$ was prevented by starving cells in media titrated to pH$_e$ 7. Gene Ontology (GO) analysis revealed that these genes are enriched for processes that are adaptive in carbon starvation, for example, fatty acid metabolism and the TCA cycle. Group 2 (n = 64) genes were not strongly induced in WT, but were inappropriately induced during starvation in *SNF5* QLC mutants and during starvation at pH$_e$ 7. GO analysis revealed that these genes are enriched for stress responses, perhaps because the failure to properly reprogram transcription leads to cellular stress. Group 3 genes (n = 51) were repressed upon carbon starvation in a pH-dependent but *SNF5* QLC-independent manner. They were repressed in all strains, but repression failed at pH$_e$ 7. Finally, group 4 genes (n = 16) were repressed in WT cells in a pH-independent manner, but failed to repress in *SNF5* QLC mutants.

We performed an analysis for the enrichment of transcription factors within the promoters of each of these gene sets using the YEASTRACT server (*Teixeira et al., 2014*). These enrichments are summarized in *Supplementary file 3*. Top hits for group 1 included the *CAT8* and *ADR1* transcription factors, which have previously been suggested to recruit the SWI/SNF complex to the ADH2 promoter (*Biddick et al., 2008b*).

In conclusion, both pH changes and the *SNF5* QLC are required for correct transcriptional reprogramming upon carbon starvation, but the dependencies are nuanced. Mutation of the *SNF5* QLC or prevention of nucleocytoplasmic acidification appears to trigger a stress response (group 2 genes). Another set of genes requires pH change for their repression upon starvation, but this pH sensing is independent of *SNF5* (group 3). A small set of genes requires the *SNF5* QLC but not pH change for repression upon starvation (group 4). Finally, a set of genes, including many of the traditionally defined 'glucose-repressed genes,' require *both* the *SNF5* QLC *and* a pH change for their induction upon carbon starvation (group 1). For these genes, point mutation of four histidines in the QLC is almost as perturbative as complete deletion of the QLC. We propose that the *SNF5* QLC senses the transient acidification that occurs upon carbon starvation to elicit transcriptional activation of this gene set. It is striking that this set is enriched for genes involved in catabolism, TCA cycle, and metabolism, given that these processes are important for energetic adaptation to acute glucose starvation.

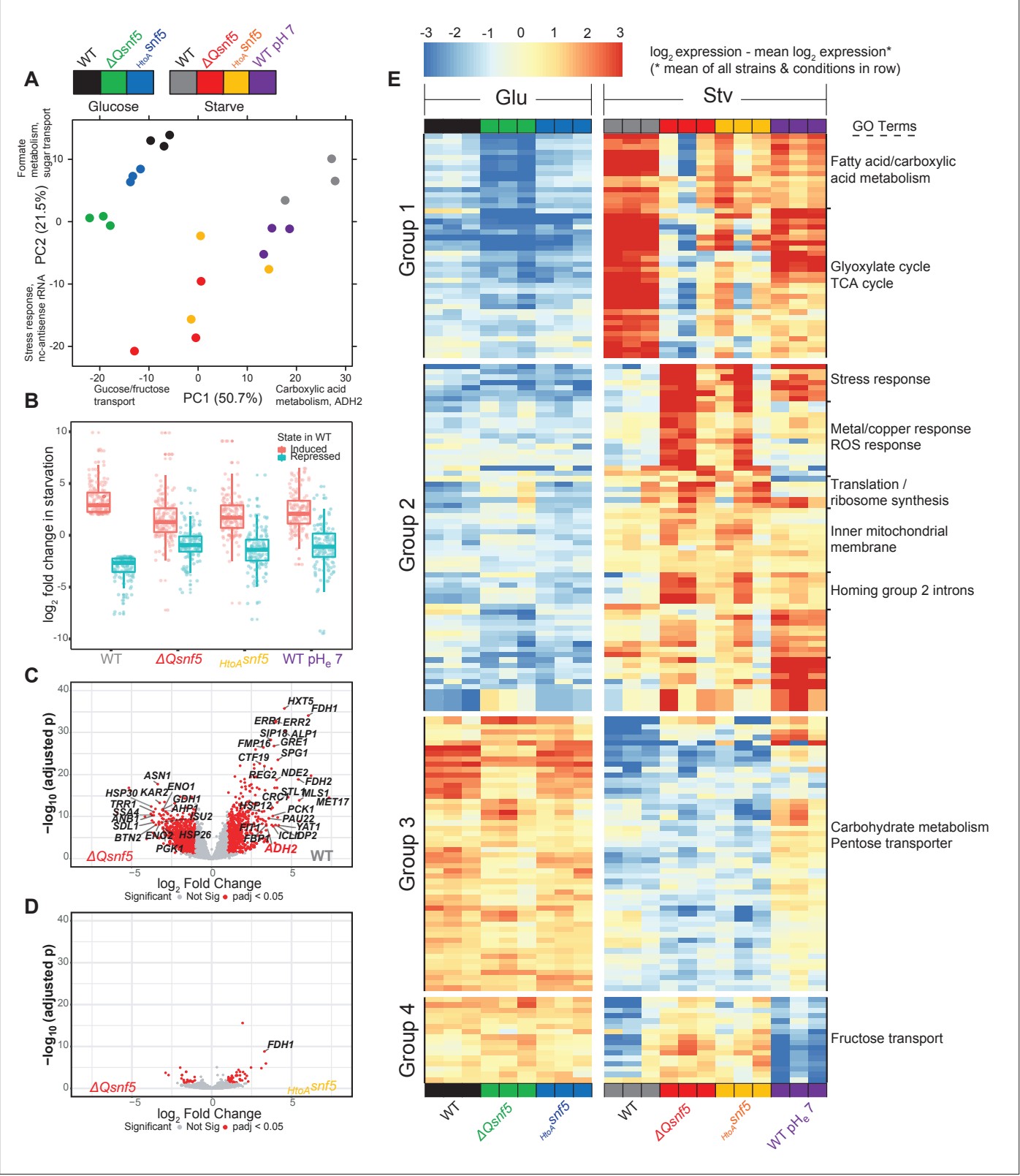

**Figure 4.** The *SNF5* QLC and acidification of the nucleocytoplasm are required for efficient widespread transcriptional reprogramming upon carbon starvation. (**A**) Principal component (PC) analysis of three RNA-seq biological replicates for each condition tested. (**B**) Expression levels of genes that were greater than threefold induced or repressed upon carbon starvation in WT strains are plotted for each *SNF5* allele. (**C**) Volcano plot showing the log₂ ratio of expression levels in WT versus *ΔQsnf5* strains (x-axis) and p-values for differential expression (y-axis). Genes with significantly different

*Figure 4 continued on next page*

*Figure 4 continued*

expression are indicated in red (log$_2$ fold change > 1 and Wald test adjusted p-value<0.05). (**D**) Volcano plot as in (**C**) but comparing expression levels in $_{HtoA}$snf5 strains to ΔQsnf5 strains. (**E**) Hierarchically clustered heat map showing expression values of 149 genes with a significant change in expression upon starvation of WT cells (log$_2$ fold change > 1 and Wald test adjusted p-value<0.05). Color code indicates gene expression relative to the mean expression of that gene across all strains and conditions, with red indicating high and blue indicating low values (see legend). Three biological replicates are shown for each experiment. Strain and condition identities are indicated at the bottom of each column. Four groups of genes with similar behavior are indicated to the left. Gene Ontology enrichment results for nine clusters of genes are shown to the right.

## The *SNF5* QLC mediates a pH-sensitive transcription factor interaction in vitro

We reasoned that pH$_i$ changes could affect the intrinsic nucleosome remodeling activity of SWI/SNF or alternatively might impact the interactions of SWI/SNF with transcription factors. Indeed, recent structural evidence (*He et al., 2021*) shows that the QLCs of not only *SNF5* but also several other SWI/SNF subunits appear to be poised for interaction with transcription factors on DNA immediately downstream of the nucleosome (*Figure 5—figure supplement 1*). We used a fluorescence-based strategy in vitro to investigate these potential pH-sensing mechanisms. A center-positioned, recombinant mononucleosome was assembled on a 200 bp DNA fragment containing a '601' nucleosome positioning sequence (*Dechassa et al., 2008*; *Figure 5A*). The nucleosomal substrate contained two binding sites for the Gal4 activator located upstream and 68 base pairs of linker DNA downstream of the nucleosome. The mononucleosome contained a Cy3 fluorophore covalently attached to the distal end of the template DNA, and Cy5 was attached to the H2A C-terminal domain. The Cy3 and Cy5 fluorophores can function as a Förster resonance energy transfer (FRET) pair only when the Cy3 donor and Cy5 acceptor are within an appropriate distance (see also *Li and Widom, 2004*). In the absence of SWI/SNF activity, the center-positioned nucleosome has a low FRET signal, but ATP-dependent mobilization of the nucleosome toward the distal DNA end leads to an increase in FRET (*Brune et al., 1994*; *Luger et al., 1999*; *Sen et al., 2017*; *Smith and Peterson, 2005*; *Zhou and Narlikar, 2016*; *Figure 5*). In the absence of competitor DNA, SWI/SNF does not require an interaction with a transcription factor to be recruited to the mononucleosome and thus intrinsic nucleosome remodeling activity can be assessed independently of recruitment. In this assay, SWI/SNF complexes containing either ΔQsnf5p or $_{HtoA}$snf5 retained full nucleosome remodeling activity (*Figure 5B–D*), as well as full DNA-stimulated ATPase activity (*Figure 5—figure supplement 2*). Furthermore, these activities were similar at pH 6.5, 7, or 7.6. Thus, we conclude that the *SNF5* QLC does not sense pH by modifying its intrinsic ATPase and nucleosome remodeling activity, at least in this in vitro context.

Next, we assessed if the *SNF5* QLC and pH changes could affect SWI/SNF interactions with transcription factors. SWI/SNF remodeling activity can be targeted to nucleosomes in vitro by Gal4 derivatives that contain acidic activation domains, an archetypal example of which is VP16 (*Yudkovsky et al., 1999*). Indeed, it was previously demonstrated that the QLC of Snf5p mediates interaction with the Gal4-VP16 transcription factor (*Prochasson et al., 2003*). To assess recruitment of SWI/SNF, we set up reactions with an excess of nonspecific competitor DNA. In these conditions, there is very little recruitment and remodeling without interaction with a transcription factor bound to the mononucleosome DNA (*Figure 5E and F*). In this context, we found that the QLC of *SNF5* was required for rapid, efficient recruitment of SWI/SNF by the Gal4-VP16 activator, and that the pH of the buffer affected this recruitment (*Figure 5F*). Within the physiological pH range (6.5–7.6), recruitment and remodeling increased with pH. This behavior might correspond to the recruitment of SWI/SNF to genes that are active at high pH$_i$ during growth in glucose. We predict that interactions with transcription factors at glucose-repressed genes would show the opposite behavior, that is, recruitment would be increased at lower pH$_i$. SWI/SNF complexes deleted for the *SNF5* QLC (containing ΔQsnf5p) had constitutively lower recruitment and were completely insensitive to pH changes over this same range (*Figure 5G*). SWI/SNF complexes containing $_{HtoA}$snf5p were even more defective that the ΔQsnf5 allele with respect to recruitment to the VP16 transcription factor (*Figure 5H*); this recruitment was barely above background levels at all pH values. Therefore, we conclude that the *SNF5* QLC can sense pH changes by modulating interactions between SWI/SNF and transcription factors. Furthermore, these results suggest that the histidines within the *SNF5 QLC* must be present and deprotonated to enable interaction with VP16.

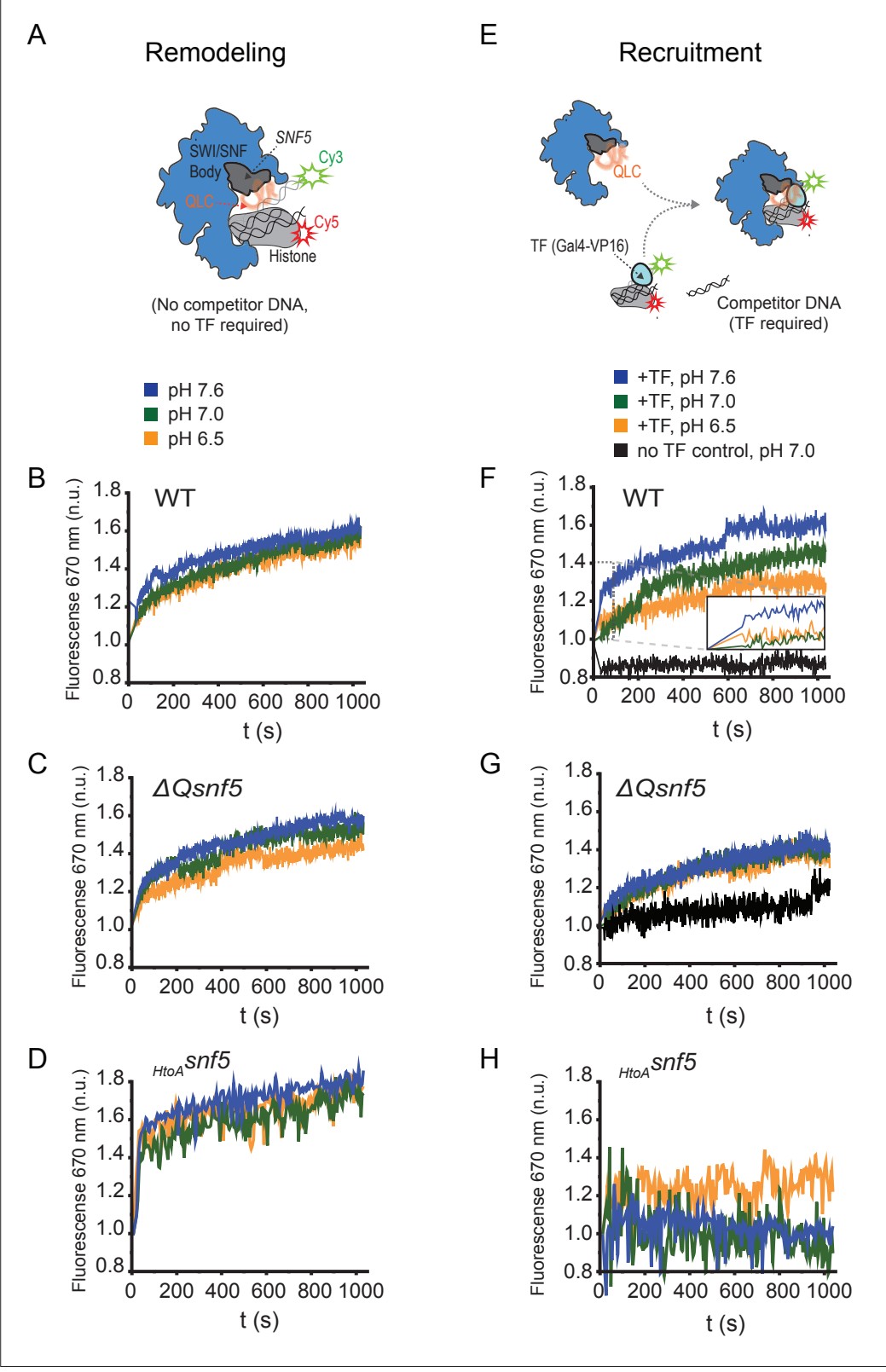

**Figure 5.** The *SNF5* QLC mediates a pH-sensitive transcription factor interaction in vitro. (**A**) Schematic: a Cy3 donor fluorophore was attached to one end of the DNA, and the histone H2A C-termini were labeled with a Cy5 acceptor fluorophore. ATP-dependent mobilization of the nucleosome to the DNA increases Förster resonance energy transfer (FRET), leading to increased emission at 670 nm. (**B**) Representative kinetic traces for

*Figure 5 continued on next page*

*Figure 5 continued*

WT (**B**), ΔQsnf5p (**C**), and _HtoA_snf5 (**D**) SWI/SNF complexes at pH 7.6 (blue), 7.0 (green), or 6.5 (orange). There is no competitor DNA, so these traces indicate intrinsic remodeling activity without requirement for recruitment by transcription factors. (**E**) Schematic: in the presence of excess competitor DNA, SWI/SNF-dependent remodeling requires recruitment by a transcription factor (Gal4-VP16). (**D**) Representative kinetic traces for WT (**F**), ΔQsnf5p (**G**), and _HtoA_snf5 (**H**) SWI/SNF complexes at pH 7.6 (blue), 7.0 (green), or 6.5 (orange). Inset on the WT panel (**F**) shows the first 100 s of the assay after ATP addition. All traces are averages of 2–4 experiments and represent FRET normalized to values prior to addition of ATP.

The online version of this article includes the following figure supplement(s) for figure 5:

**Figure supplement 1.** QLCs of SWI/SNF cluster around putative transcription factor interaction sites, as do low-complexity sequences of human BAF complex.

**Figure supplement 2.** Basal ATPase activity is not affected by pH, and Förster resonance energy transfer (FRET) changes require ATP hydrolysis.

## Protonation of histidines leads to conformational expansion of the *SNF5* QLC

How might pH change be sensed by *SNF5*? As described above (*Figure 1B*), glutamine-rich low-complexity sequences (QLCs) are enriched for histidines, and they are also depleted for charged amino acids (*Figure 1B*). Charged amino acids have repeatedly been shown to govern the conformational behavior of disordered regions (*Mao et al., 2010*; *Müller-Späth et al., 2010*; *Sorensen and Kjaergaard, 2019*). Given that histidine protonation alters the local charge density of a sequence, we hypothesized that the charge-depleted QLCs may be poised to undergo protonation-dependent changes in conformational behavior. To test this idea, we performed all-atom Monte Carlo simulations to assess the conformational ensemble of a 50 amino acid region of the *SNF5* QLC (residues 71–120) that contained three histidines, two of which we had mutated to alanine in our experiments (*Figure 6A*). We performed simulations with histidines in both uncharged and protonated states to mimic possible charges of this polypeptide at the pH found in the nucleocytoplasm in glucose and carbon starvation, respectively. These simulations generated ensembles of almost 50,000 distinct conformations (representative images shown in *Figure 6B*). To quantify conformational changes, we examined the radius of gyration, a metric that describes the global dimensions of a disordered region (*Figure 6C*). Protonation of the wild-type sequence led to a striking increase in the radius of gyration, driven by intramolecular electrostatic repulsions (*Figure 6D*, left). In contrast, when 2/3 histidines were replaced with alanines, no such change was observed (*Figure 6D*, right). For context, we also calculated an apparent scaling exponent ($v^{app}$), a dimensionless parameter that can also be used to quantify chain dimensions. This analysis showed that protonation of the wild-type sequence led to a change in $v^{app}$ from 0.48 to 0.55, comparable to the magnitude of changes observed in previous studies of mutations that fundamentally altered intermolecular interactions in other low-complexity disordered regions (*Martin et al., 2020*; *Sorensen and Kjaergaard, 2019*). These results suggest that small changes in sequence charge density can elicit a relatively large change in conformational behavior. An analogous (albeit less pronounced) effect was observed for the second QLC subregion that we mutated (residues 195–233) (*Figure 6—figure supplement 1*). Taken together, our results suggest that charge-depleted disordered regions (such as QLCs) are poised to undergo pH-dependent conformational rearrangement. This inference offers the beginnings of a mechanism for pH sensing by SWI/SNF: the conformational expansion of the QLC sequence upon nucleocytoplasmic acidification may tune the propensity for SWI/SNF to interact with transcription factors (*Figure 6E*).

## Discussion

Intracellular pH changes occur in many physiological contexts, including cell cycle progression (*Gagliardi and Shain, 2013*), the circadian rhythm of crassulacean acid metabolism plants (*Hafke et al., 2001*), oxidative stress (*van Schalkwyk et al., 2013*), heat shock (*Triandafillou et al., 2020*), osmotic stress (*Karagiannis and Young, 2001*), and changes in nutritional state (*Jacquel et al., 2020*; *Orij et al., 2009*). However, the physiological role of these pH_i fluctuations and the molecular mechanisms to detect them remain poorly understood. Prior results have emphasized the inactivation of processes in response to cytosolic acidification (*Joyner et al., 2016*; *Munder et al., 2016*; *Petrovska*

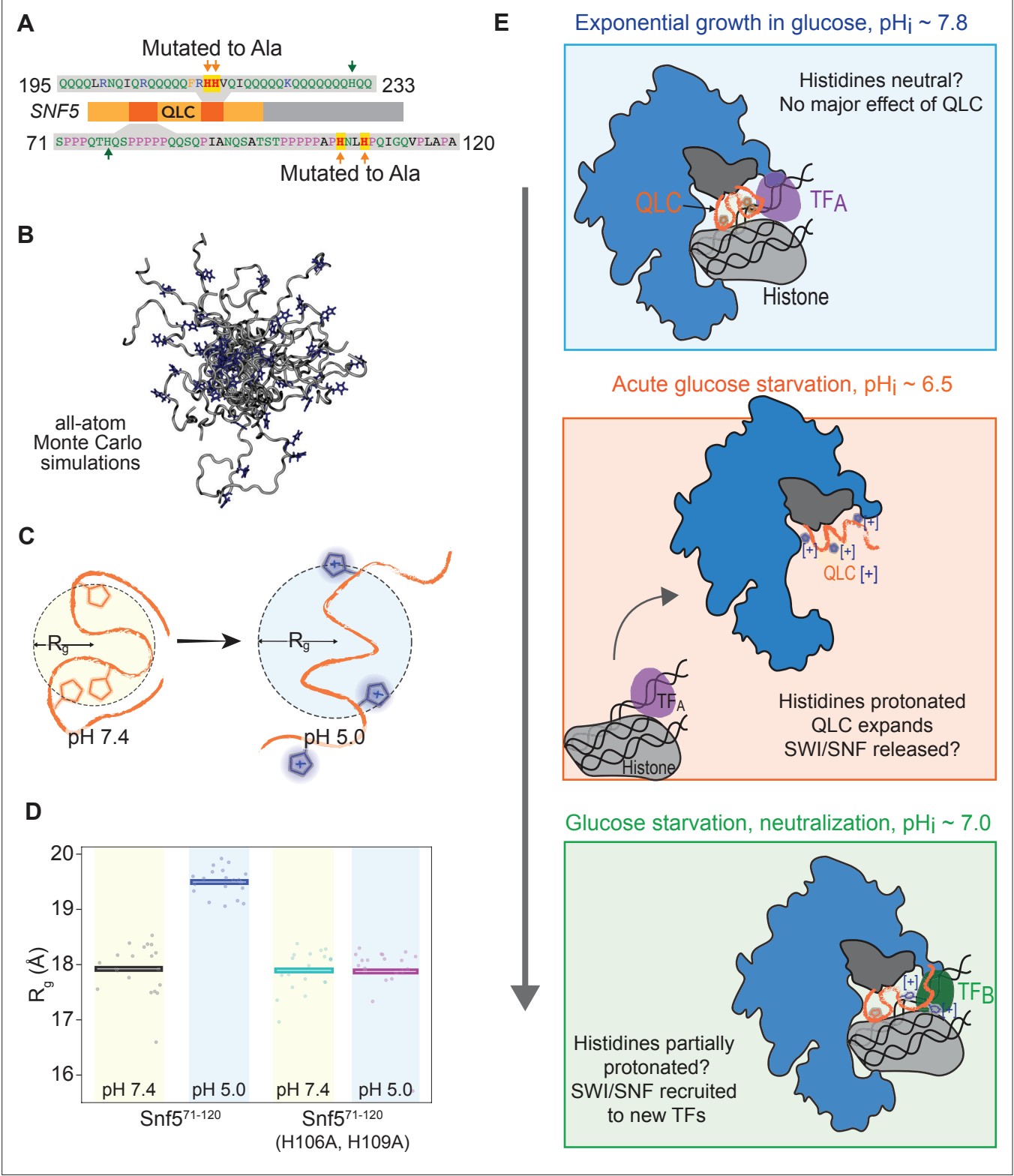

**Figure 6.** Protonation of histidines leads to conformational expansion of the *SNF5* QLC. (**A**) Schematic of the *SNF5* gene (center) with the N-terminal QLC in orange and the two simulated peptides in dark orange. Sequences of the simulated peptides and identities of histidines mutated in both the *HtoA snf5* yeast strain and in simulations are indicated. (**B**) Representative images of conformations sampled in Monte Carlo all-atom simulations. (**C**) Cartoon depicting quantification of radius of gyration ($R_g$). (**D**) Radius of gyration ($R_g$, y-axis) of simulations of amino acids 71–120 of the *SNF5* QLC with histidines either neutral (pH 7.4) or protonated (pH 5.0). Left two datasets are for the native peptide, right two datasets are with 2/3 histidines (H106 and

*Figure 6 continued on next page*

*Figure 6 continued*

H109) replaced with alanine, mimicking the $_{HtoA}snf5$ allele. Points represent the mean $R_g$ from all conformations sampled in each independent simulation (beginning from distinct random initial conformers). Bars represent the mean values of all simulations. (**E**) Model of SWI/SNF regulation during carbon starvation. (Top) In glucose (pH$_i$ ~ 7.8), the *SNF5* QLC is unprotonated. SWI/SNF is engaged by transcription factors that prevent transcription of glucose repressed genes or that activate other genes (TF$_A$). (Middle) Upon acute carbon starvation, pH$_i$ drops to ~6.5, leading to protonation of histidines in the *SNF5* QLC. Conformational expansion of the QLC may aid the release of SWI/SNF from some transcription factors (TF$_A$) and potentially drive recruitment to others (not shown). (Bottom) As the cell adapts to carbon starvation, pH$_i$ neutralizes to ~7.0. Histidines within the *SNF5* QLC may be partially protonated? The pK$_a$ of histidine is highly context-dependent. The QLC may aid recruitment of SWI/SNF to the promoters of glucose-repressed genes, thus leading to their expression.

The online version of this article includes the following figure supplement(s) for figure 6:

**Figure supplement 1.** A second peptide within the N-terminal QLC of *SNF5* undergoes conformational expansion upon protonation.

*et al., 2014*). However, it is unclear how necessary modifications to the cell can occur if cellular dynamics are uniformly decreased. Much less has been reported regarding a potential role of fluctuations in pH$_i$ as a signal to activate specific cellular programs. In this work, we found that transient acidification is required for activation of glucose-repressed genes. Therefore, our work establishes a positive regulatory role for nucleocytoplasmic pH changes during carbon starvation.

Previous studies of intracellular state during glucose starvation based on population averages reported a simple decrease in pH$_i$ (*Orij et al., 2009*). In this work, we used single-cell measurements of both pH$_i$ and gene expression, and found that two coexisting subpopulations arose upon acute glucose starvation, one with pH$_i$ ~ 5.5 and a second at ~6.5. The latter population recovered to neutral pH$_i$ and then induced glucose-repressed genes, while the former remained dormant in an acidified state. We have not yet determined the mechanism that drives the bifurcation in pH response. It is possible that this bistability provides a form of bet-hedging (*Levy et al., 2012*) where some cells attempt to respond to carbon starvation, while others enter a dormant state (*Munder et al., 2016*). However, we have yet to discover any condition where the population with lower pH$_i$ and delayed transcriptional activation has an advantage. An alternative explanation is that these cells are failing to correctly adapt to starvation, perhaps undergoing a metabolic crisis, as suggested in a recent study (*Jacquel et al., 2020*).

It is becoming clear that intracellular pH is an important mechanism of biological control. It was previously shown that the protonation state of phosphatidic acid (PA) determines binding to the transcription factor Opi1, coupling membrane biogenesis and intracellular pH (*Young et al., 2010*). We focused our studies on the N-terminal region of *SNF5* because it is known to be important for the response to carbon starvation and contains a large low-complexity region enriched in both glutamine and histidine residues. Histidines are good candidates for pH sensors as they can change protonation state over the recorded range of physiological pH fluctuations, and their pK$_a$ can be tuned substantially depending on local sequence context. Consistent with this hypothesis, we found that the *SNF5* QLC and the histidines embedded within were required for transcriptional reprogramming.

Our in vitro assays showed that the intrinsic ATPase and nucleosome remodeling activities of SWI/SNF are robust to pH changes from 6.5 to 7.6. However, recruitment of the SWI/SNF complex by a model transcription factor (GAL4-VP16) was pH-sensitive, and this pH dependence was dependent on both the *SNF5* QLC and the four central histidines within this domain. In this case, the recruitment by GAL4-VP16 was inhibited at pH 6.5. We speculate that low pH$_i$ favors release of SWI/SNF from activators that it is bound to in glucose conditions, and then the subsequent partial recovery in pH$_i$ could allow it to bind to a different set of activators (e.g., *ADR1* and *CAT8*), thus recruiting it to genes that are expressed during starvation. This model is consistent with the requirement for both acidification and subsequent neutralization for expression of *ADH2* (*Figure 3*). In principle, the conformational dynamics of the *SNF5* QLC could be distinct at all three stages (*Figure 6E*). There are almost certainly additional pH-sensing elements of the transcriptional machinery that also take part in this reprogramming; multiple candidates are present among the of transcription factors that were enriched in our RNA-seq experiments (*Supplementary file 3*).

Low-complexity sequences, including QLCs, tend to be intrinsically disordered and therefore highly solvent exposed. A recent large-scale study of intrinsically disordered sequences showed that their conformational behavior is inherently sensitive to changes in their solution environment (*Holehouse and Sukenik, 2020*; *Moses et al., 2020*). Similarly, our simulations revealed that histidine

protonation may lead the *SNF5* QLC to expand dramatically. This provides a potential mechanism for pH sensing: upon acidification, histidines become positively charged, leading QLCs to adopt a more expanded state, perhaps revealing short linear interaction motifs (SLIMs), reducing the entropic cost of binding to interaction partners, preventing polar-mediated protein-protein interactions, or facilitating electrostatic-mediated contacts. The enrichment of histidines in QLCs hints that this could be a general, widespread mechanism to regulate cell biology in response to pH changes.

Glutamine-rich low-complexity sequences have been predominantly studied in the context of disease. Nine neurodegenerative illnesses, including Huntington's disease, are thought to be caused by neurotoxic aggregation seeded by proteins that contain polyglutamines created by expansion of CAG trinucleotide repeats (*Fan et al., 2014*). However, polyglutamines and glutamine-rich sequences are relatively abundant in *Eukaryotic* cells: more than 100 human proteins contain QLCs, and the *Dictyostelium* and *Drosophilid* phyla have QLCs in ~10% and ~5% of their proteins, respectively (*Schaefer et al., 2012*). Furthermore, there is clear evidence of purifying selection to maintain polyQs in the *Drosophilids* (*Huntley and Clark, 2007*). This prevalence and conservation suggest an important biological function for these sequences. Recent work in *Ashbya gossypii* has revealed a role for QLC-containing proteins in the organization of the cytoplasm through phase separation into liquid droplets to enable subcellular localization of signaling molecules (*Zhang et al., 2015*). More generally, polyglutamine has been shown to drive self-association into a variety of higher-order assemblies, from fibrils to nanoscopic spheres to liquid droplets (*Crick et al., 2013*; *Peskett et al., 2018*; *Posey et al., 2018*). Taken together, these results imply that QLCs may offer a general mechanism to drive protein-protein interactions. In this study, we have identified a role for QLCs in the SWI/SNF complex as pH sensors. Our current model (*Figure 6E*) is that the *SNF5* QLC partakes in heterotypic protein interactions that are modulated by protonation of histidines when the cell interior acidifies. However, we do not rule out the possibility for homotypic interactions and higher-order assembly of multiple SWI/SNF complexes.

All cells must modify gene expression to respond to environmental changes. This phenotypic plasticity is essential to all life, from single-celled organisms fighting to thrive in an ever-changing environment, to the complex genomic reprogramming that must occur during development and tissue homeostasis in plants and animals. Despite the differences between these organisms, the mechanisms that regulate gene expression are highly conserved. Changes in intracellular pH are increasingly emerging as a signal through which life perceives and reacts to its environment. This work provides a new role for glutamine-rich low-complexity sequences as molecular sensors for these pH changes.

## Materials and methods

**Key resources table**

| Reagent type (species) or resource | Designation | Source or reference | Identifiers | Additional information |
|---|---|---|---|---|
| Gene (*Saccharomyces cerevisiae*) | *SNF5* | https://www.yeastgenome.org/ | SGD:S000000493 | |
| Gene (*S. cerevisiae*) | *SNF2* | https://www.yeastgenome.org/ | SGD:S000005816 | |
| Gene (pHluorin) | pHluorin | doi:10.1099/mic.0.022038-0 | | |
| Strain, strain background (*S. cerevisiae* S288c) | BY4741 | doi:https://doi.org/10.1002/(SICI)1097-0061 (19980130)14:2<115::AID-YEA204>3.0.CO;2-2 | | All strains used in this study are derived form BY4741 |
| Other | LH3647 | *ADH2::P_{ADH2}-mCherry-URA3 snf2::SNF2-TAP-His3MX6* | | Yeast strain used to purify SWI/SNF complex |
| Other | LH3649 | *ΔQsnf5-HIS3 ADH2::P_{ADH2}-mCherry-URA3 snf2::SNF2-TAP-kanMX6* | | Yeast strain used to purify SWI/SNF complex containing *ΔQsnf5* |

*Continued on next page*

*Continued*

| Reagent type (species) or resource | Designation | Source or reference | Identifiers | Additional information |
|---|---|---|---|---|
| Other | LH3652 | $_{HtoA}$snf5-HIS3 ADH2::P$_{ADH2}$-mCherry-URA3 snf2::SNF2-TAP-kanMX6 | | Yeast strain used to purify SWI/SNF complex containing $_{HtoA}$snf5 |
| Recombinant DNA reagent | Plasmid (pRS316) | GenBank: U03442 | | Used to complement SNF5 gene in snf5Δ strains prior to removal using 5FOA |
| Recombinant DNA reagent | Plasmid (pRS306) | GenBank: U03438 | | SNF5 and snf5 mutant alleles were all cloned into pRS306 and pRS303 |
| Recombinant DNA reagent | Plasmid (pRS303) | GenBank: U03435 | | SNF5 and snf5 mutant alleles were all cloned into pRS306 and pRS303 |
| Antibody | Rabbit polyclonal IgG | Sigma | Cat# 12-370 | |
| Antibody | Fluorescently labeled goat anti-rabbit polyclonal | LI-COR Biosciences | Cat# 926-68071 | Western blot (1:15,000 dilution) |
| Antibody | Rabbit polyclonal anti-glucokinase | US Biological | Cat# H2035-01 | Western blot (1:3000 dilution) |
| Antibody | Fluorescently labeled goat anti-rabbit polyclonal | LI-COR Biosciences | Cat# 926-32211 | Western blot (1:15,000 dilution) |

## Yeast strains used in this study

All strains were derived from LH2145.

| Strain | Genotype |
|---|---|
| LH2145 | WT, Mat a from sporulation of BY4743: ura3Δ0 his3Δ0 leu22Δ0 met15Δ0 |
| LH2090 | ΔQsnf5::kanMX6 |
| LH2971 | SNF5-TAP-His3MX6 |
| LH2973 | ΔQsnf5-TAP-His3MX6 |
| LH2974 | $_{HtoA}$snf5-HIS3 |
| LH2975 | $_{HtoA}$snf5-TAP-kanMX6 |
| LH2991 | ADH2::P$_{ADH2}$-mCherry-URA3 |
| LH2992 | ΔQsnf5-kanMX6 ADH2::P$_{ADH2}$-mCherry-URA3 |
| LH2993 | $_{HtoA}$snf5-HIS3 ADH2::P$_{ADH2}$-mCherry-URA3 |
| LH3486 | met15Δ0 SNF5::kanMX6 (CEN/ARS-SNF5::URA3) |
| LH3513 | snf5Δ::kanMX6 ADH2::P$_{ADH2}$-mCherry-URA3 (CEN/ARS-SNF5::URA3) |
| LH3632 | snf5Δ::kanMX6 ADH2::P$_{ADH2}$-mCherry-URA3 TRP1::pHluorin-natMX6 (CEN/ARS-SNF5::URA3) |
| LH3647 | ADH2::P$_{ADH2}$-mCherry-URA3 snf2::SNF2-TAP-His3MX6 |
| LH3649 | ΔQsnf5-HIS3 ADH2::P$_{ADH2}$-mCherry-URA3 snf2::SNF2-TAP-kanMX6 |
| LH3652 | $_{HtoA}$snf5-HIS3 ADH2::P$_{ADH2}$-mCherry-URA3 snf2::SNF2-TAP-kanMX6 |
| LH3705 | SNF5 ADH2::P$_{ADH2}$-mCherry-URA3 leu2::pHluorin-LEU2 |

*Continued on next page*

*Continued*

| Strain | Genotype |
|--------|----------|
| LH3707 | *ΔQsnf5::kanMX6 ADH2::P*<sub>ADH2</sub>*-mCherry-URA3 leu2::pHluorin-LEU2* |
| LH3713 | <sub>HtoA</sub>*snf5-HIS3 ADH2::P*<sub>ADH2</sub>*-mCherry-URA3 leu2::pHluorin-LEU2* |

## Plasmids used in this study

| Plasmid | Identity |
|---------|----------|
| pLH226 | pFA6a- *ΔQsnf5*-GFP(S65T)-KANMX6 |
| pLH416 | pFA6a-*SNF5*-GFP-KANMX6 |
| pLH887 | pRS316-SNF5 (CEN/ARS plasmid) |
| pLH931 | pFA6a-<sub>4HtoA</sub>*snf5*-KANMX6 |
| pLH963 | pFA6a-*SNF5*-TAP-KANMX6 |
| pLH964 | pFA6a-*SNF5*-TAP-HIS3MX6 |
| pLH998 | pRS306-*P*<sub>ADH2</sub>-*mCherry* |
| pLH1085 | pFA6a-<sub>6HtoA</sub>*snf5*-HIS3MX6 |
| pLH1093 | pFA6a-*3'snf2*-TAP-KANMX6 |
| pLH1097 | pRS305-*P*<sub>TDH3</sub>-pHluorin |
| pLH1206 | pFA6a-*3'snf2*-TAP-NATMX |

## Cloning and yeast transformations

Yeast strains used in this study were all in the S288c strain background (derived from BY4743). The sequences of all genes in this study were obtained from the *S. cerevisiae* genome database (http://www.yeastgenome.org/).

We cloned the various *SNF5* alleles into plasmids from the Longtine/Pringle collection (*Longtine et al., 1998*). We assembled plasmids by PCR or gene synthesis (IDT gene blocks) followed by Gibson cloning (*Gibson et al., 2009*). Then, plasmids were linearized and used to overwrite the endogenous locus by sigma homologous recombination using homology to both ends of the target gene.

The *ΔQsnf5* gene lacks the N-terminal 282 amino acids that comprise a glutamine-rich low-complexity domain. Methionine 283 serves as the ATG for the *ΔQ-SNF5* gene. In the <sub>HtoA</sub>*snf5* allele, histidines 106, 109, 213, and 214 were replaced by alanine using mutagenic primers to amplify three fragments of the QLC region, which were combined by Gibson assembly into an *SNF5* parent plasmid linearized with BamH1 and Sac1.

We noticed that the slow growth null strain phenotype of the *snf5Δ* was partially lost over time, presumably due to suppressor mutations. Therefore, to avoid these spontaneous suppressors, we first introduced a CEN/ARS plasmid carrying the *SNF5* gene under its own promoter and the *URA3* auxotrophic selection marker. Then, a kanMX6 resistance cassette, amplified with primers with homology at the 5′ and 3′ of the *SNF5* gene, was used to delete the entire chromosomal *SNF5* ORF by homologous recombination. We subsequently cured strains of the CEN/ARS plasmid carrying WT *SNF5* by negative selection against its URA3 locus by streaking for single colonies on 5-FOA plates immediately before each experiment to analyze the *snf5Δ* phenotype.

The *P*<sub>ADH2</sub>-*mCherry* reporter was cloned into integrating pRS collection plasmids (*Chee and Haase, 2012*). *URA3* (pRS306) or *LEU2* (pRS305) were used as auxotrophic selection markers. The 835 base pairs upstream of the *ADH2* gene were used as the promoter (*P*<sub>ADH2</sub>). *P*<sub>ADH2</sub> and the mCherry ORF were amplified by PCR and assembled into linearized pRS plasmids (Sac1/Asc1) by Gibson assembly. These plasmids were cut in the middle of the *ADH2* promoter using the Sph1 restriction endonuclease and integrated into the endogenous *ADH2* locus by homologous recombination.

The *pHluorin* gene was also cloned into integrating pRS collection plasmids. *URA3* (pRS306) and *LEU2* (pRS305) were used for selection. The plasmid with the *pHluorin* gene was obtained as described

in *Orij et al., 2009*. We amplified the *pHluorin* gene and the strong *TDH3* promoter and used Gibson assembly to clone these fragments into pRS plasmids linearized with Sac1 and Asc1. Another strategy was to clone the *pHluorin* gene and a natMX6 cassette into the integrating pRS304 plasmid (that contains *TRP1*), which was then linearized within the *TRP1* cassette using HindIII and integrated into the *TRP1* locus.

A C-terminal TAP tag was used to visualize Snf5 and Snf2 proteins in Western blots. pRS plasmids were used, but the cloning strategy was slightly different. A 3′ fragment of the *SNF5* and SNF2 genes was PCR amplified without the Stop codon. This segment does not contain a promoter or an ATG codon for translation initiation. The TAP tag was then amplified by PCR and cloned together with the 3′ of *SNF5* and *SNF2* ORFs by Gibson assembly into pRS plasmids with linearized Sac1 and Asc1. Plasmids were linearized in the 3′ of the *SNF5* or *SNF2* ORFs with StuI and XbaI, respectively, to linearize the plasmid allowing integration it into the 3′ of each gene locus by homologous recombination. Therefore, transformation results in a functional promoter at the endogenous locus fused to the TAP tag.

The *SNF5-GFP* strain was obtained from the yeast GFP collection (*Huh et al., 2003*), a gift of the Drubin/Barnes laboratory at UC Berkeley. The *SNF2-GFP* fused strain was made by the same strategy used for the TAP tagged strain above.

*Supplementary files 6 and 7* list strains and plasmids generated in this study.

## Culture media

Most experiments, unless indicated, were performed in synthetic complete (SC) media (13.4 g/L yeast nitrogen base and ammonium sulfate; 2 g/L amino acid mix and 2% glucose). Carbon starvation media was SC media without glucose, supplemented with sorbitol, a nonfermentable carbon source to avoid osmotic shock during glucose starvation (6.7 g/L YNB + ammonium sulfate; 2 g/L amino acid mix and 100 mM sorbitol). The pH of starvation media ($pH_e$) was adjusted using NaOH.

## Growth assays

Growth rates were determined in an Infinite M200 plate reader (Tecan) in 96-well microtiter plates using 200 μL total volume, cultured at 30°C and agitated at 800 rpm. Cells were pre-cultured overnight to log-phase (or subjected to other indicated pre-culture conditions) and then seeded at an A600 of 0.1 (based on a path length of ~0.3 cm) in SC media with various carbon sources. All measurements were performed in triplicate.

## Glucose starvation

Cultures were incubated in a rotating incubator at 30°C and grown overnight (14–16 hr) to an OD between 0.2 and 0.3. Note that it is extremely important to prevent culture OD from exceeding 0.3, and results are different if cells are allowed to saturate and then diluted back. Thus, it is imperative to grow cultures from colonies on plates for >16 hr without ever exceeding OD 0.3 to obtain reproducible results. Typically, we would inoculate 3 mL cultures and make a series of 4–5 1/5 dilutions of this starting culture to be sure to catch an appropriate culture the following day. 3 mL of OD 0.2–0.3 culture were centrifuged at 6000 rpm for 3 min and resuspended in 3 mL starvation media (SC sorbitol at various $pH_e$). This spin and resuspension was repeated two more times to ensure complete removal of glucose. Finally, cells were resuspended in 3 mL of starvation media. For flow cytometry, 200 μL samples were transferred to a well of a 96-well plate at each time point. During the course of time-lapse experiments, culture aliquots were set aside at 4°C. An LSR II flow cytometer with an HTS automated sampler was used for all measurements. 10,000 cells were analyzed at each time point.

## Nucleocytoplasmic pH measurements

Nucleocytoplasmic pH ($pH_i$) was measured by flow cytometry or microscopy. The ratiometric, pH-sensitive GFP variant, *pHluorin*, was used to measure pH based on the ratio of fluorescence from two excitation wavelengths. The settings used for LSR II flow cytometer were AmCyan (excitation 457, emission 491) and FITC (excitation 494, emission 520). AmCyan emission increases with pH, while FITC emission decreases. A calibration curve was made for each strain in each experiment. To generate a calibration curve, glycolysis and respiration were poisoned using 2-deoxyglucose and azide. This treatment leads to a complete loss of cellular ATP, and the nucleocytoplasmic pH equilibrates to the

extracellular pH. We used the calibration buffers published by Patricia Kane's group (*Diakov et al., 2013*): 50 mM MES (2-(N-morpholino) ethanesulfonic acid), 50 mM HEPES (4-(2-hydroxyethyl)-1-piper azineethanesulfonic acid), 50 mM KCl, 50 mM NaCl, 0.2 M ammonium acetate, 10 mM sodium azide, 10 mM 2-deoxyglucose. Buffers were titrated to the desired pH with HCl or NaOH. Sodium azide and 2-deoxyglucose were always added fresh.

## RT-qPCR

For qPCR and RNA-seq, RNA was extracted with the 'High pure RNA isolation kit' (Roche) following the manufacturer's instructions. Three biological replicates were performed. cDNAs and qPCR were made with iSCRIPT and iTAQ universal SYBR green supermix by Bio-Rad, following the manufacturer's instructions. Samples processed were exponentially growing culture (+Glu) or acute glucose star-vation for 4 hr in media titrated to pH 5.5 or 7.5. Primers for qPCR were taken from *Biddick et al., 2008a*; for *ADH2* and *FBP1* genes: forward (GTC TAT CTC CAT TGT CGG CTC), reverse (GCC CTT CTC CAT CTT TTC GTA), and forward (CTT TCT CGG CTA GGT ATG TTG G), reverse (ACC TCA GTT TTC CGT TGG G). *ACT1* was used as an internal control; primers were: forward (TGG ATT CCG GTG ATG GTG TT), reverse (TCA AAA TGG CGT GAG GTA GAG A).

## RNA-sequencing

We performed RNA-sequencing analysis to determine the extent of the requirement for the *SNF5* QLC in the activation of glucose-repressed genes. Three biological replicates were performed. Total RNA was extracted from WT, *ΔQ-snf5,* and $_{HtoA}$snf5 strains during exponential growth (+Glu) and after 4 hr of acute glucose starvation. In addition, WT strains were acutely starved in media titrated to pH 7. Next, poly-A selection was performed using Dynabeads and libraries were performed following the manufacturer's indications. Sequencing of the 32 samples was performed on an Illu-mina HiSeq on two lanes. RNA-seq data were aligned to the University of California, Santa Cruz (UCSC), sacCer2 genome using Kallisto (0.43.0, http://www.nature.com/nbt/journal/v34/n5/full/nbt.3519.html) and downstream visualization and analysis was in R (3.2.2). Differential gene expres-sion analysis, heat maps, and volcano plots were created using DESeq2. A Wald test was used to determine differentially expressed genes. Euclidean distance was used to calculate clustering for heat maps, with some manual curation to remove small clusters with no significant GO hits, and to consolidate clusters that had similar behavior. RNA-seq R-code can be found at https://github.com/gbritt/SWI_SNF_pH_Sensor_RNASeq., (copy archived at swh:1:rev:802f3d233210c-02c66b745e414a6f7aa1385e379). RNA-seq datasets are deposited at GEO accession number GSE174687 (available here).

## Western blots

Strains containing *SNF5* and *SNF2* fused to the TAP tag were used. Given the low concentration of these proteins, they were extracted with trichloroacetic acid (TCA): 3 mL culture was pelleted by centrifugation for 2 min at 6000 rpm and then frozen in liquid nitrogen. Pellets were thawed on ice and resuspended in 200 µL of 20% TCA, ~0.4 g of glass beads were added to each tube. Samples were lysed by bead beating four times for 2 min with 2 min of resting in ice in each cycle. Supernatants were extracted using a total of 1 mL of 5% TCA and precipitated for 20 min at 14,000 rpm at 4°C. Finally, pellets were resuspended in 212 µL of Laemmli sample buffer and pH adjusted with ~26 µL of Tris buffer pH 8. Samples were run on 7–12% gradient polyacrylamide gels with Thermo Fisher PageRuler prestained protein ladder 10–18 kDa. Proteins were transferred to a nitrocellulose membrane, which was then blocked with 5% nonfat milk and incubated with a rabbit IgG primary antibody (which binds to the protein A moiety of the TAP tag) for 1 hr and then with fluorescently labeled goat anti-rabbit secondary antibody IRDye 680RD goat-anti-rabbit (LI-COR Biosciences, Cat# 926-68071, 1:15,000 dilution). Anti-glucokinase was used as a loading control (rabbit-anti-Hxk1, US Biological, Cat# H2035-01, RRID:AB_2629457, Salem, MA, 1:3,000 dilution) followed by IRDye 800CW goat-anti-rabbit (LI-COR Biosciences, Cat# 926-32211, 1:15,000 dilution). Membranes were visualized using a LI-COR Odyssey CLx scanner with Image Studio 3.1 software. Fluorescence emission was quantified at 700 and 800 nM.

## Co-immunoprecipitation of SWI/SNF complex

To evaluate the assembly state of the SWI/SNF complex, we immunoprecipitated Snf2p. To enable this experiment, we constructed strains in which the *SNF2* gene was tagged at the C-terminus with a TAP tag (*Puig et al., 2001*). For each purification, 6 L of cells were grown in YPD to an OD of 1.2. Cells were broken open using glass beads in buffer A (40 mM HEPES [K+], pH 7.5, 10% glycerol, 350 mM KCl, 0.1% Tween-20, supplemented with 20 µg/mL leupeptin, 20 µg/mL pepstatin, 1 µg/mL benzamidine hydrochloride, and 100 µM PMSF) using a BioSpec bead beater followed by treatment with 75 units of benzonase for 20 min (to digest nucleic acids). Heparin was added to a final concentration of 10 µg/mL. The extract was clarified by first spinning at 15,000 rpm in a SS34 Sorvall rotor for 30 min at 4°C, followed by centrifugation at 45,000 rpm for 1.5 hr at 4°C in a Beckman ultracentrifuge. The soluble extract was incubated with IgG Sepharose beads for 4 hr at 4°C using gentle rotation. IgG Sepharose bound proteins were washed five times in buffer A and once in buffer B (10 mM Tris-HCl, pH 8.0, 10% glycerol, 150 mM NaCl, 0.5 mM EDTA, 0.1% NP40, 1 mM DTT, supplemented with 20 µg/mL leupeptin, 20 µg/mL pepstatin, 1 µg/mL benzamidine hydrochloride, and 100 µM PMSF). Bound protein complexes were incubated in buffer B with TEV protease overnight at 4°C using gentle rotation. The eluted protein was collected, $CaCl_2$ was added to a final concentration of 2 mM and bound to Calmodulin Sepharose beads for 4 hr at 4°C using gentle rotation. Following binding, the protein-bound Calmodulin Sepharose beads were washed five times in buffer C (10 mM Tris-HCl, pH 8.0, 10% glycerol, 150 mM KCl, 2 mM $CaCl_2$, 0.1% NP40, 1 mM DTT, supplemented with 20 µg/mL leupeptin, 20 µg/mL pepstatin, 1 µg/mL benzamidine hydrochloride, and 100 µM PMSF). The bound proteins were eluted in buffer D 10 mM Tris-HCl, pH 8.0, 10% glycerol, 150 mM KCl, 2 mM EGTA, 0.1% NP40, 0.5 mM DTT, supplemented with 20 µg/mL leupeptin, 20 µg/mL pepstatin, 1 µg/mL benzamidine hydrochloride, and 100 µM PMSF. The protein complexes were resolved by SDS-PAGE and visualized by silver staining.

## Chromatin immunoprecipitation of SWI/SNF

For ChIP of the SWI/SNF complex, we constructed strains in which the *SNF2* gene was tagged at the C-terminus with a TAP tag, as above. $1.25 \times 10^8$ cells were collected for each mutant and condition and fixed on 1% formaldehyde for 20 min to crosslink proteins to chromatin, and then the reaction was stopped with 136 mM glycine. Cells were pelleted and frozen in liquid nitrogen. Cells were then resuspended in 400 µL lysis buffer (0.1% deoxycholic acid, 1 mM EDTA, 50 mM HEPES pH 7.5, 140 mM NaCl, 1% Triton X-100, and 5 mM phenanthroline), mixed with 400 µL glass beads, and then lysed by vortexing for 15 min. The same lysis buffer was used to rinse the glass beads once more to recover remaining lysate. Lysates were then sonicated for 10 s, six times in ice to sheer chromatin, and then incubated with 40 µL of IgG-conjugated magnetic beads per sample ($1 \times 10^8$ beads) and incubated for 24 hr at 4°C on a nutator (Dynabeads m-270 epoxy [Thermo Fisher 14301] conjugated to IgG from rabbit serum [Sigma-Aldrich I5006]; for conjugation protocol, see here).

After binding, samples were washed once with 600 µL buffer 2 (0.1 % deoxycholic acid, 1 mM EDTA, 50 mM HEPES pH 7.5, 500 mM NaCl, 1% Triton X-100, and 5 mM phenanthroline) and then washed once with 600 µL buffer 3 (0.5% deoxycholic acid, 1 mM EDTA, 250 mM LiCl, 0.5% NP-50, 10 mM Tris pH 7.9, and 5 mM phenanthroline), and finally washed once with 600 µL buffer TE.

The crosslinking between DNA and proteins was reversed by heating in elution buffer (50 mM Tris-HCl pH 7.5, 10 mM EDTA, and 1% SDS) for 2 hr at 42°C and then for 8 hr at 65°C. Eluted DNA was purified using QIAGEN kit (28104) according to the manufacturer's instructions.

qPCR was performed using a Roche LightCycler 480 SYBR green master mix (04707516001) following the manufacturer's instructions.

Two sets of primers were used to amplify for *ADH2* (*Parua et al., 2014*):

> 1F: ACC ATC CAC TTC ACG AGA CTG A, 1R:AAA AGT CGC TAC TGG CAC TC
> 2F: GAG TGC CAG TAG CGA CTT TTT, 2R: ACT TGC CGT TGG ATT CGT AG

## Data fitting

Fluorescence intensity from the $P_{ADH2}$-*mCherry* reporter and ratiometric fluorescence measurements from pHluorin were fit with a single or double Gaussian curve for statistical analysis using MATLAB (MathWorks). The choice of a single or double Gaussian fit was determined by assessing which fit gave

the least residuals. For simplicity, the height (mode) of each Gaussian peak was used to determine the fraction of cells in each population rather than the area because peaks overlapped in many conditions.

## Sequence analysis of QLCs

### Identification of QLCs

QLCs were defined as subregions of the proteome that have an average fraction of glutamine residues of 25% or higher (minimum fraction), the maximum interruption between any two glutamine residues is less than 17 residues, and the whole QLC is at least 15 residues in length (minimum length) (*Figure 1—figure supplement 2A*). All XLCs (low-complexity subsequences for all amino acids, including glutamine) are provided online for further exploration and analysis (see GitHub). Secondly, systematic variation of the maximum interruption size to ask how the number of QLCs and number of residues found revealed that 17 residues was the value that maximized the number of QLCs and the number of residues found within QLCs, offering an optimally permissive value under the 0.25 or greater fraction of glutamine threshold.

### Computation of per-residue conservation

Per-residue conservation was calculated by taking orthologous fungal proteins from the yeast genome order browser, aligning those using Clustal Omega, and calculating the Jensen–Shannon divergence as implemented by Caprah and Singh using the BLOSUM62 matrix (*Byrne and Wolfe, 2005*; *Capra and Singh, 2007*; *Henikoff and Henikoff, 1992*; *Lin, 1991*; *Sievers et al., 2011*).

### Proteome-wide analysis

*S. cerevisiae*, *Dictyostelium*, *Drosophila*, and human proteins were obtained from UniProt. Sequence analysis was performed with SHEPHARD (https://shephard.readthedocs.io/). Predicted disorder scores, IDR identification, and predicted pLDDT scores were performed by metapredict (*Emenecker et al., 2021*; ). QLCs and full proteomes are provided at here.

### Proteome-wide per-residue enrichment or depletion in QLCs

To compute the enrichment or depletion of specific amino acid residues in QLCs, we determined the fraction of non-glutamine residues in QLCs compared to the fraction of non-glutamine residues across the entire proteome. Specifically, for each proteome (*S. cerevisiae*, *D. discoideum*, *D. melanogaster*, and *Homo sapiens*) we first computed the proteome-wide background by taking the complete set of all protein sequences, removing all glutamine residues from all proteins, and then computing the fraction of the proteome made up of the remaining 19 amino acids. For each proteome, we then identified the full set of QLCs and repeated the analysis. The $\log_2$ of the ratio of the fraction of each amino acid in a QLC vs. across the proteome was used to compute enrichment or depletion for different amino acids within QLCs.

### Proteome-wide per-residue enrichment or depletion of QLCs with respect to all XLCs

To compute enrichment of different amino acids in QLCs compared to other low-complexity domains (XLCs), we repeated the analysis above using XLCs defined by enrichment for non-glutamine residues, and then re-computed non-glutamine enrichment as was done for the whole proteome. The complete set of all XLC subsequences for all four proteomes is provided.

## Nucleosome remodeling assays

### SWI/SNF purification

SWI/SNF complexes were purified from yeast strains with a TAP protocol as previously described (*Smith and Peterson, 2005*). Cells were grown in YPAD media and harvested at $OD_{600}$ = 3, and flash frozen and stored at –80°C. Yeast cells were lysed using a cryomill (PM100 Retsch). Ground cell powder was resuspended in E buffer (20 mM HEPES, 350 mM NaCl, 0.1% Tween-20, 10% glycerol, pH 7.5), with fresh 1 mM DTT and protease inhibitors (0.1 mg/mL phenylmethylsulfonyl fluoride, 2 µg/mL leupeptin, 2 µg/mL pepstatin, 1 mM benzamidine) and incubated on ice for 30 min. The crude lysate was clarified first by centrifugation 3K rpm for 15 min, and then 40K rpm for 60 min at 4°C. The clear

lysate was transferred to a 250 mL Falcon tube and incubated with 400 µL IgG resin slurry (washed previously with E buffer without protease inhibitors) for 2 hr at 4°C. The resin was washed extensively with E buffer and protease inhibitors, and the protein-bound resin was incubated with 300 units TEV protease overnight at 4°C. The eluent was collected, incubated with 400 µL calmodulin affinity resin, washed previously with E buffer with fresh protease inhibitors, DTT and 2 mM $CaCl_2$, for 2 hr at 4°C. Resin washed with the same buffer and SWI/SNF was eluted with E buffer with protease inhibitors, DTT, and 10 mM EGTA. The eluent was dialyzed in E buffer with PMSF, DTT, and 50 µM $ZnCl_2$ at least three times. The dialyzed protein was concentrated with a Vivaspin column, aliquoted, flash frozen, and kept at –80°C. SWI/SNF concentration was quantified by electrophoresis on 10% SDS-PAGE gel alongside a BSA standard titration, followed by SYPRO Ruby (Thermo Fisher Scientific) staining overnight and using ImageQuant 1D gel analysis.

## Mononucleosome reconstitutions

Recombinant octamers were reconstructed from isolated histones as described previously (*Luger et al., 1999*). In summary, recombinant human H2A (K125C), H2B, and H3 histones and *Xenopus laevis* H4 were isolated from *Escherichia coli* (Rosetta 2 [DE3] with and without pLysS). In order to label human H2A, a cysteine mutation was introduced at residue K125 via site-directed mutagenesis, which was labeled with Cy5 fluorophore attached to maleimide group (*Zhou and Narlikar, 2016*). DNA fragments were generated from 601 nucleosome positioning sequence and 2x Gal4 recognition sites with primers purchased from IDT. For FRET experiments, PCR amplification of labeled DNA fragments was as follows: 500 nM Cy3 labeled (5′-Cy3/TCCCCAGTCACGACGTTGTAAAAC-3′) and unlabeled primers (5′-ACCATGATTACGCCAAGCTTCGG-3′), 200 µM dNTPs, 0.1 ng/µL p159-2xGal4 plasmid kindly donated by Blaine Bartholomew, 0.02 U/µL NEB Phusion DNA polymerase, 1× Phusion High Fidelity Buffer. For ATPase assays, two unlabeled primers used (PrimerW: 5′-GTACCCGGGGAT CCTCTAGAGTG-3′, PrimerS: 5′-GATCCTAATGACCAAGGAAAGCA-3′) under same PCR conditions with NEB Taq DNA Polymerase with 1× NEB ThermoPol Buffer. 400 nM fluorescently labeled and unlabeled mononucleosomes were reconstituted via salt gradient at 4°C with a peristaltic pump as described previously (*Luger et al., 1999*), with 600 mL high salt buffer (10 mM Tris-HCl, pH 7.4, 1 mM EDTA, 2 M KCl, 1 mM DTT) exchanged with 3 L of low salt buffer (10 mM Tris-HCl, pH 7.4, 1 mM EDTA, 50 mM KCl, 1 mM DTT) over 20 hr. The quality of the nucleosomes was checked by visualizing proteins on a 5% native-PAGE gel and scanning fluorescence ratios of labeled nucleosomes on an ISS PC1 spectrofluorometer.

## FRET-based nucleosome remodeling

The fluorescence resonance energy transfer between Cy3-labeled DNA and Cy5-labeled octamer was used to measure the remodeling and recruitment activity of SWI/SNF using an ISS PC1 spectrofluorometer. The remodeling activity was measured by the increase in FRET signal in that occurred as a consequence of nucleosome sliding the DNA template. The reaction was performed under three different pH conditions: pH 6.5 (25 mM MES, 0.2 mM EDTA, 5 mM $MgCl_2$, 70 mM KCl, 1 mM DTT), pH 7 (25 mM Tris, 0.2 mM EDTA, 5 mM $MgCl_2$, 70 mM KCl, 1 mM DTT), and pH 7.6 (25 mM HEPES, 0.2 mM EDTA, 5 mM $MgCl_2$, 70 mM KCl, 1 mM DTT). Remodeling reactions contained 2 nM or 4 nM (WT or mutant) SWI/SNF, 5 nM nucleosome, and 100 µM ATP or AMP-PNP. A 100 s pre-scan of the reaction was taken before the reaction started and the time-dependent fluorescence measurements started after addition of ATP or AMP-PNP for 1000s at room temperature. Similarly, recruitment assays were performed in three different buffer conditions: pH 6.5, pH 7, and pH 7.6. The recruitment assays contained 2 nM or 4 nM (WT or mutant) SWI/SNF, 5 nM nucleosome, 4 nM competitor DNA, 100 µM Gal4–VP16 (Protein One, P1019-02) and 100 µM ATP or AMP-PNP, together with respective controls (*Sen et al., 2017*). 100 s of pre-scans and 1000s of time-dependent enzyme kinetics were measured. At least 2–4 kinetic traces were collected per reaction. Data were normalized to their respective pre-scans to account for variation between reactions. The time-dependent FRET signals were excited at 530 nm and measured at 670 nm. The data analysis was performed using the OriginLab software package.

## ATPase activity measurements

7-Diethylamino-3-[N-(2-maleimidoethyl)-carbamoyl]-coumarin-conjugated phosphate binding protein A197C (MDCC-PBP) (*Brune et al., 1994*) was used to detect inorganic phosphate ($P_i$) release from ATPase activity in real time. Before the reaction, ATP was cleared of free $P_i$ by performing a mopping reaction. In order to mop the ATP, 10 mM ATP was incubated with 1 U/mL PNPase (Sigma, N2415-100UN) and 200 µM 7-methylguanosine (Sigma, M0627-100MG) in mopping buffer (25 mM HEPES, 75 mM NaCl, 5 mM MgCl$_2$, 1 mM DTT) for 2 hr at room temperature. ATPase assay reaction conditions were 2 nM SWI/SNF, 5 nM nucleosome, and 100 µM ATP in respective pH buffers; pH 6.5 (25 mM MES, 0.2 mM EDTA, 5 mM MgCl$_2$, 70 mM KCl, 1 mM DTT), pH 7 (25 mM Tris, 0.2 mM EDTA, 5 mM MgCl$_2$, 70 mM KCl, 1 mM DTT), or pH 7.6 (25 mM HEPES, 0.2 mM EDTA, 5 mM MgCl$_2$, 70 mM KCl, 1 mM DTT). The measurements were performed on a Tecan Infinite 1000, with excitation at 405 nm and emission at 460 nm. Pre-scan measurements were taken to detect the basal level of signal per reaction. The time-dependent measurements were taken after starting the reaction by ATP addition. At least 3–4 kinetic traces were analyzed using the steady-state equation using GraphPad Prism 8 software.

## All-atom simulations

All-atom simulations were run with the ABSINTH implicit solvent model and CAMPARI Monte Carlo simulation (V2.0; http://campari.sourceforge.net/; *Vitalis and Pappu, 2009*). The combination of ABSINTH and CAMPARI has been used to examine the conformational behavior of disordered proteins with good agreement to experiment (*Cubuk et al., 2020*; *Fuertes et al., 2017*; *Martin et al., 2020*).

All simulations were started from randomly generated nonoverlapping random-coil conformations, with each independent simulations using a unique starting structure. Monte Carlo simulations perturb and evolve the system via a series of moves that alter backbone and sidechain dihedral angles, as well as rigid-body coordinates of both protein sequences and explicit ions. Simulation analysis was performed using CAMPARITraj (http://www.ctraj.com/) and MDTraj (*McGibbon et al., 2015*).

ABSINTH simulations were performed with the ion parameters derived by Mao et al. and using the abs_opls_3.4.prm parameters (*Mao et al., 2010*). All simulations were run at 15 mM NaCl and 325 K, a simulation temperature previously shown to be a good proxy for bona fide ambient temperature (*Das et al., 2016*; *Martin et al., 2020*). A summary of the simulation input details is provided in *Supplementary file 5*. For SNF5[71-120] simulations, 20 independent simulations were run for each combination of pH (as defined by histidine protonation state) and mutational state. For SNF5[195-223], the high glutamine content made conformational sampling challenging, as has been observed in previous glutamine-rich systems, reflecting the tendency for polyglutamine to undergo intramolecular chain collapse (*Crick et al., 2006*; *Newcombe et al., 2018*; *Warner et al., 2017*). To address this challenge, we ran hundreds of short simulations (with a longer equilibration period than in SNF[71-120]) that are guaranteed to be uncorrelated due to their complete independence (*Vitalis and Caflisch, 2010*). Simulation code and details can be found at https://github.com/holehouse-lab/supportingdata/tree/master/2021/Gutierrez_QLC_2021.

## Bioinformatic analysis

All protein sequence analyses were performed with localCIDER, with FASTA files read by protfasta (https://github.com/holehouse-lab/protfasta; *Holehouse et al., 2017*; *Holehouse, 2021*). Sequence alignments were performed using Clustal Omega (*Sievers et al., 2011*). Sequence conservation was computed using default properties in with the score_conservation program as defined by *Capra and Singh, 2007*. Proteomes were downloaded from *UniProt Consortium, 2015*.

Low-complexity sequences were identified using Wootton-Federhen complexity (*Ginell and Holehouse, 2020*; *Wootton and Federhen, 1993*). Sequence complexity is calculated over a sliding window size of 15 residues, and a threshold of 0.6 was used for binary classification of a residue as 'low' or 'high' complexity. After an initial sweep, gaps of up to three 'high-complexity residues' between regions of low-complexity residues were converted to low-complexity. Finally, contiguous stretches of 30 residues or longer were taken as the complete set of low-complexity regions in the proteome. The full set of those SEG-defined LCDs for human, *Drosophila*, *Dictyostelium*, and *Cerevisiae* proteomes is provided as FASTA files available here.

## Acknowledgements

We thank Conor Howard for help with initial bioinformatics and conception of this project and Morgan Delarue for help with MATLAB analysis. We thank David Truong, Sudarshan Pinglay, and JoAnna Klein for help in writing the manuscript; Ivan Tarride for help with figure design; and Karsten Weis, Jeremy Thorner, and Douglas Koshland for advice, strains, plasmids, and reagents. We thank Cindy Hernandez for help with growth curves. We gratefully acknowledge funding from the William Bowes Fellows program, the Vilcek Foundation, the HHMI HCIA summer institute, NIH R01 GM132447 and R37 CA240765, the American Cancer Society Cornelia T Bailey Foundation Research Scholar Grant, RSG-19-073-01-TBE, and the Pershing Square Sohn Cancer Research Award (LJH); Becas Chile (JIG); and the National Science Foundation Graduate Research Fellows Program (GB).

## Additional information

### Funding

| Funder | Grant reference number | Author |
| --- | --- | --- |
| Becas Chile | | J Ignacio Gutierrez |
| National Science Foundation | Graduate Research Fellows Program | Gregory P Brittingham |
| Pershing Square Sohn Cancer Research Award | | Liam J Holt |
| National Cancer Institute | R37 CA240765 | Liam J Holt |
| National Institute of General Medical Sciences | R01 GM132447 | Liam J Holt |
| American Cancer Society Cornelia T. Bailey Foundation Research Scholar Grant | RSG-19-073-01-TBE | Liam J Holt |

The funders had no role in study design, data collection and interpretation, or the decision to submit the work for publication.

### Author contributions

J Ignacio Gutierrez, Conceptualization, Formal analysis, Investigation, Methodology, Writing – original draft, Writing – review and editing; Gregory P Brittingham, Conceptualization, Data curation, Formal analysis, Investigation, Methodology, Software, Validation, Visualization, Writing – review and editing; Yonca Karadeniz, Kathleen D Tran, Formal analysis, Investigation, Methodology; Arnob Dutta, Formal analysis, Investigation, Methodology, Writing – review and editing; Alex S Holehouse, Conceptualization, Formal analysis, Investigation, Methodology, Resources, Software, Visualization, Writing – review and editing; Craig L Peterson, Conceptualization, Formal analysis, Funding acquisition, Investigation, Supervision, Writing – review and editing; Liam J Holt, Conceptualization, Funding acquisition, Investigation, Methodology, Project administration, Visualization, Writing – original draft, Writing – review and editing

### Author ORCIDs

J Ignacio Gutierrez http://orcid.org/0000-0002-9017-8384
Yonca Karadeniz http://orcid.org/0000-0002-8299-551X
Alex S Holehouse http://orcid.org/0000-0002-4155-5729
Liam J Holt http://orcid.org/0000-0002-4002-0861

### Decision letter and Author response

Decision letter https://doi.org/10.7554/eLife.70344.sa1
Author response https://doi.org/10.7554/eLife.70344.sa2

## Additional files

### Supplementary files
• Supplementary file 1. Sequences of glutamine-rich low-complexity sequences (QLCs) in the *Saccharomyces cerevisiae* genome. All *S. cerevisiae* QLCs identified using the parameters optimized in *Figure 1—figure supplement 2* are included in this summary table.

• Supplementary file 2. Comparison of sequence properties of *SNF5* N-terminal intrinsically disordered regions (IDRs). Comparison of the IDRs of *SNF5* orthologues from *Ascomycete* fungi, with the number of glutamines and histidines indicated.

• Supplementary file 3. Transcription factors enriched in each gene group from RNA-seq analysis. The YEASTRACT server used to find transcription factors enriched within the promoters of each of four gene sets defined by hierarchical clustering of genes significantly regulated upon carbon starvation (see *Figure 4E*). YEASTRACT search settings were DNA binding plus expression evidence; TF acting as either activator or inhibitor.

• Supplementary file 4. SNF5 subregions examined by all-atom Monte Carlo simulations.

• Supplementary file 5. Parameters used for all-atom Monte Carlo simulations.

• Supplementary file 6. Yeast strains used in this study.

• Supplementary file 7. Plasmids used in this study.

• Transparent reporting form

### Data availability
Simulation code and details can be found at: https://github.com/holehouse-lab/supportingdata/tree/master/2021/Gutierrez_QLC_2021, (copy archived at swh:1:rev:bafdd4e42c496ecdf1da134c-9ca5ac709b273ae5; path=/2021/Gutierrez_QLC_2021/) RNA-seq R-code can be found at: https://github.com/gbritt/SWI_SNF_pH_Sensor_RNASeq, (copy archived at swh:1:rev:802f3d233210c-02c66b745e414a6f7aa1385e379) datasets are deposited at GEO accession number GSE174687 https://www.ncbi.nlm.nih.gov/geo/query/acc.cgi?acc=GSE174687.

The following datasets were generated:

| Author(s) | Year | Dataset title | Dataset URL | Database and Identifier |
|---|---|---|---|---|
| Gutierrez JI | 2021 | Simulation code and details | https://github.com/holehouse-lab/supportingdata/tree/master/2021/Gutierrez_QLC_2021 | GitHub, GitHub |
| Brittingham GP | 2021 | RNA-seq R-code | https://github.com/gbritt/SWI_SNF_pH_Sensor_RNASeq | GitHub, GitHub |
| Brittingham GP, Holt LJ, Gutierrez JI | 2021 | SWI/SNF senses carbon starvation with a pH-sensitive low complexity sequence | https://www.ncbi.nlm.nih.gov/geo/query/acc.cgi?acc=GSE174687 | NCBI Gene Expression Omnibus, GSE174687 |

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
