## [Editor Report]

This study has considerable merit in providing evidence that the Q-rich low-complexity domain in Snf5, and the histidine residues located therein, functions as a sensor of the drop in intracellular pH that accompanies glucose starvation to mediate SWI/SNF recruitment and transcriptional activation of the battery of genes derepressed under these conditions in order to reprogram carbon utilization. The work is multifaceted in combining yeast genetics, single-cell assays of gene expression and intracellular pH, genome-wide analysis of gene expression changes by RNA-seq, and in vitro biophysical analysis of activator-dependent SWI/SNF recruitment and nucleosome remodeling in a purified system.

---

## [Decision Letter]

[Editors' note: this paper was reviewed by Review Commons.]

**Decision letter after peer review:**

Thank you for submitting your article "SWI/SNF senses carbon starvation with a pH-sensitive low complexity sequence" for consideration by *eLife*. We have evaluated the three reviews from Reviews Common and your rebuttal, and also carefully read the manuscript ourselves. The Reviewing Editor has drafted this to help you prepare a revised submission.

Essential revisions:

1. In addition to making all of the proposed revisions and additions of new analysis described in the authors' rebuttal letter to the three Review Commons reviews, the following additional experiments are required:

2. As requested by more than one of the referees (#1 and #2), it is necessary to extend the in vitro analysis of Figure 5 to include the HtoA variant of Snf5. As the authors noted, the N-terminal domain of Snf5 was shown previously to interact with the VP16 activation domain. It is critical to show that the histidine residues in this domain are key to this interaction using their elegant in vitro assay for SWI/SNF recruitment. This request does not involve simply "investigating more subtle alleles", as the effect of the HtoA variant is central to their model; and it does not involve developing a new reconstitution assay-at odds with the main arguments in the authors' rebuttal for declining to conduct this key requested experiment.

3. It is necessary to demonstrate by ChIP assay that recruitment of SWI/SNF to the key target gene ADH2 in response to carbon starvation is impaired by the Snf5 HtoA mutation and by an external pH of 7 in WT cells to provide in vivo evidence that protonation of the His residues in the Snf5 QLC stimulates SWI/SNF recruitment to a target promoter activated during glucose starvation. This is particularly important because recruitment of SWI/SNF by Gal4-VP16 was lower at pH6.5 versus pH7.0 in the in vitro recruitment assay of Figure 5, and thus does not mimic the effect of pH on expression of ADH2 and other genes induced by glucose starvation in yeast cells, as was noted by Ref. #1. This requested ChIP experiment falls far short of the ChIP-seq and ATAC-sec or MNase ChIP-seq experiments requested by Ref. #3 in terms of time and expense, while providing essential support for the proposed mechanistic model.

---

## [Author Response]

Essential revisions:1. In addition to making all of the proposed revisions and additions of new analysis described in the authors' rebuttal letter to the three Review Commons reviews, the following additional experiments are required:2. As requested by more than one of the referees (#1 and #2), it is necessary to extend the in vitro analysis of Figure 5 to include the HtoA variant of Snf5. As the authors noted, the N-terminal domain of Snf5 was shown previously to interact with the VP16 activation domain. It is critical to show that the histidine residues in this domain are key to this interaction using their elegant in vitro assay for SWI/SNF recruitment. This request does not involve simply "investigating more subtle alleles", as the effect of the HtoA variant is central to their model; and it does not involve developing a new reconstitution assay-at odds with the main arguments in the authors' rebuttal for declining to conduct this key requested experiment.

We have completed these additional in vitro experiments. We found that the SWI/SNF complex containing the HtoA *Snf5p* variant was fully functional in terms of basal chromatin remodeling activity (new figure 5D) but is completely defective in terms of recruitment to the VP16 transcription activation domain (new figure 5H.) We interpret this result to mean that the *Snf5* QLC requires unprotonated histidines to be present to allow interaction with VP16, and that alanines are not able to substitute for this interaction. This experiment confirms the importance of the four central histidines in the *Snf5* QLC for interaction with transcription factors.

3. It is necessary to demonstrate by ChIP assay that recruitment of SWI/SNF to the key target gene ADH2 in response to carbon starvation is impaired by the Snf5 HtoA mutation and by an external pH of 7 in WT cells to provide in vivo evidence that protonation of the His residues in the Snf5 QLC stimulates SWI/SNF recruitment to a target promoter activated during glucose starvation. This is particularly important because recruitment of SWI/SNF by Gal4-VP16 was lower at pH6.5 versus pH7.0 in the in vitro recruitment assay of Figure 5, and thus does not mimic the effect of pH on expression of ADH2 and other genes induced by glucose starvation in yeast cells, as was noted by Ref. #1. This requested ChIP experiment falls far short of the ChIP-seq and ATAC-sec or MNase ChIP-seq experiments requested by Ref. #3 in terms of time and expense, while providing essential support for the proposed mechanistic model.

We have completed these additional ChIP experiments (new Figure 1—figure supplement 8). We find that recruitment of SWI/SNF to the *ADH2* promoter is impaired in *ΔQsnf5* and *_HtoA_snf5* strains and when extracellular pH is buffered to 7.5. These results are consistent with our hypothesis that protonation of the *SNF5* QLC is important for recruitment to this glucose-repressed promoter.

[Editors' note: we include below the reviews that the authors received from Review Commons, along with the authors’ responses.]

Major experiments and analyses added to the new manuscript in response to Review Commons Reviews:

Regarding bioinformatic analyses of amino-acid enrichment in glutamine-rich low-complexity sequences (QLCs) and of the evolution of low-complexity sequences in *SWI/SNF.*

– The operational definition of QLCs was somewhat arbitrary. We have updated our analysis using an optimized definitions of QLC to better understand amino-acid enrichments throughout evolution. The parameter optimization is show in new Figure 1—figure supplement 2.

– We previously only explored the Snf5 QLC in the *Ascomycota*. In the revised manuscript, we expanded our analysis to explore *Eukaryotic* diversity more broadly. These results are presented in the new Figure 1—figure supplement 5. These results indicate that the *SNF5* QLC was gained in the fungal lineage that includes the *Basidiomycetes* and *Ascomycota.*

Regarding the differences between SWI/SNF and human BAF complex.

– We only really talked about human BAF in one paragraph of the discussion. We have completely deleted this paragraph in the current version. We have also added an annotated diagram comparing the human BAF and yeast SWI/SNF complexes indicating the positions of the QLCs and other major low complexity sequences – new Figure 5 —figure supplement 1.

Responses to individual reviews:

Reviewer #1 (Evidence, reproducibility and clarity (Required)):Gutiérrez et al. report an intriguing set of experimental results on the transcriptional role of SWI/SNF subunit Snf5 in pH sensing in the context of carbon starvation. More generally, the authors propose that histidine residues embedded in polyglutamine stretches sense pH and mechanistically transduce this as protein conformational changes that,in their system, affect TF interactions with polyglutamine stretches present in coTF subunits.Overall the data are solid and the paper is very well written. Still, the discussion is a little lengthy, including emphasis on extrapolation of the yeast results to human cells, which reads more like a grant proposal than a discussion.

We are happy to reduce the discussion. We have deleted the paragraph that speculates regarding relevance to human BAF complex in cancer.

Although somewhat speculative, the conclusions of the authors are supported by high-quality data. I therefore support publication of a revised manuscript.

We thank the reviewer.

Minor:– Mediator – ADR1 interactions are independent of SWI/SNF according to PMID: 18250152 which is consistent with the author's observations. Could this aspect of ADH2 transcriptional regulation be discussed?

Our interpretation of this study is that ADH2 induction becomes independent of SWI/SNF only if ADR1 is artificially fused to a mediator subunit. The results of the study are consistent with our model.

– What about RNA stability to explain the results? PMID: 26667037

This study shows that elements in the ADH2 promoter influence mRNA stability, particularly leading to destabilization of the transcript in the presence of glucose. We don’t explore the inactivation of ADH2 in glucose here, but it could be interesting to think about in the future.

– In relation to the speculations of the authors in the discussion; do yeast recover their pH if put back into 2% glucose? Could cell sorting be applied to study the short and long-term fate of the low and hi pH subpopulations.

We have sorted cells to see if there are any differences in fate for the high and low pH subpopulations. We have found that the high pH cells (cells that recover pH) are always better, in terms of fitness and heart stress tolerance, than the low pH cells. We have added discussion of these data (new Figure 2 supplement 2).

– The RNAseq data should be deposited in a public database (GEO or SRA).

We have deposited the data in the GEO database with accession number GSE174687.

Figures:Figure 2C, 3B black vs grey lines? It is not immediately obvious from the legends of Figures 2 and 3 that the grey and black lines concern the low and high pHi subpopulation.

We have clarified this point in the legend.

Figure 5: 5B shows that the DeltaQsnf5 SWI/SNF complex is pH sensitive, even though it lacks the proposed pH sensor domain? 5D I would have expected the low pH to favor Gal4-VP16 recruitment, if this reconstitution of TF-mediated recruitment mimics the situation at ADH2. The result is the opposite. This apparently paradoxical result is only addressed in the discussion, but is somewhat misleading in the results parts. Importantly, why did the authors not assay the 4HtoASNF5 version of SWI/SNF in vitro?

In figure 5B (now figure 5C), there is no significant effect of pH on the basal rate of nucleosome remodeling for either WT or dQ-snf5.

In figure 5D (now figure 5F), we don’t think this is a paradox. We believe that the change in structure of the SNF5 QLC favors interactions with some transcription factors and disfavors others. We have addressed this point in the Results section to improve readability.

We performed the assay with the 4HtoA allele (figure 5D and H). We found that the intrinsic activity of this complex was equal to WT and (similar to WT) unaffected by pH (figure 5D). Interestingly, the 4HtoA was even more defective that the dQ allele with respect to recruitment to the VP16 transcription factor (figure 5H). This recruitment was barely above background levels at all pH values*.*

“development and tissue homeostasis in plants and metazoa.” > plants and animals.

We made this change to improve readability.

…“to its environment. This work provides a new role for glutamine-rich low-complexity sequences as molecular sensors for these pH changes.” > histidine-bearing Q-rich low complexity sequences.…. Would be more accurate phrasing of the proposed mechanistic model.

We made this change to improve accuracy.

Reviewer #3 (Significance (Required)):The authors propose a novel mechanism of dbTF-coTF interaction modulation (via intracellular pH modulation of His proteonation) and provide evidence to support it.The major shortcoming of this study concerns the reconstitution of pH-dependent TF-mediated recruitment of the 4HtoASnf5-SWI/SNF complex in vitro. In fact, reconstitution of SNF5-ADR1 interaction might be the best experiment, since Gal4-VP16 behaves opposite to the prediction of the model. Alternatively, Gal4-AH could be used?

We have added the 4HtoA allele to figure 5 (see response above).

Gal4-AH is predicted to behave just like VP16 – they are both considered to be classical acidic activators. We have thought about Adr1, but no one has defined the activation domain, thus making this experiment currently infeasible. We have added discussion of the model that pH allows SWI/SNF to reassort itself to other activators, like Adr1.

Referee Cross-commentingThe editor letter could include the suggestion by one of the reviewers to include rolox – vitamin E in their remodeling reaction. Journal of Fluorescence volume 17, pages785-795 (2007) Single-Pair FRET Microscopy Reveals Mononucleosome Dynamics.

This is an anti-oxidant used in single molecule FRET studies to eliminate photo-blinking events. This is not necessary for ensemble FRET, as the timescale is longer (blinking doesn’t matter) and we look at the entire population. This approach is not commonly used in ensemble FRET.

I would like to support Major Revision as a decision.In my opinion, reviewer 3 very correctly cast the results in a physiological perspective when saying "Their model is.… After a transient acidification, the conformational expansion is reversed leading to interaction with different transcription factors and ultimately to an altered transcriptional response." The genetic involvement of the histidines in Snf5 appears to be solid even though more details can be provided by the authors. To my knowledge, the molecular mechanism has not yet been proposed, and therefore, has value as a hypothesis, even if it is restricted to fungi.We all agree that the reconstitution experiments with the 4 histidine versus wt complex is missing and showing pH titration results would be great (reviewer 2).Aren't those good reasons to ask whether the authors are in a position to provide more experimental data leading to a major revision?

We have added the pH titration curve (new Figure 2 supplement 1).

We have further explored the bioinformatics across the broader Eukaryotic tree.

This analysis suggests that the SNF5 QLC regulatory module was gained in Ascomycota.

Reviewer #2 (Evidence, reproducibility and clarity (Required)):Summary:This paper addresses how yeast cells adapt to carbon starvation, a process that has long been known to depend on gene regulation by the SWI/SNF chromatin remodeling complex. The authors find that a Qrich low complexity sequence (QLC) in the SNF5 subunit of SWI/SNF, and His residues within this sequence, are required for the response to glucose starvation. Sequence analysis indicates that QLCs with an enrichment of His are common in many eukaryotes. Using a fluorescent ADH2 reporter to track the transcription response to glucose starvation, and a fluorescent pH reporter. Single cell measurements of ADH2 induction and intercellular pH during carbon starvation at different external pH conditions led the authors to conclude that the signal for induction of ADH2 is the transient intracellular acidification that occurs in response to glucose starvation. SNF5 QLC is required both for normal kinetics of transient acidification and for additional steps during ADH2 induction. RNA-seq of WT and SNF5 QLC mutants at different conditions identified 4 groups of genes with different regulatory patterns. Bioinformatic analyses showed that genes involved in acute glucose-starvation response are pH and SNF5 QLC-dependent. in vitro nucleosome remodeling assays were used to demonstrate that neither the SNF5 QLC nor pH (over the relevant range) regulate ATPase or nucleosome remodelling activity. However, using a remodeling assay that depends on transcription factor recruitment, wild type SWI/SNF, but not SWI/SNF with the SNF5 QLC deleted, shows decreased recruitment at lower pH. All atom Monte Carlo simulations predict that segments of the QLC undergo protonation dependent conformational expansion, which is suggested to form the mechanistic basis of pH-sensitive transcription factor interactions. Sequence analysis demonstrates that QLCs with His residues are conserved in SWI5 across many fungi, and that QLCs are and more particular histidines in these low complexity regions, were proposed to play an evolutionarily conserved role in many eukaryotes based on in silico sequence analyses.The conclusions of the authors are mostly well supported by their experimental findings. Starting with the in silico sequence analyses that indicate a broad evolutionary relevance of QLCs and their histidines, the authors move on to dissect QLC function at multiple scales. Growth assays in combination with the pADH2 fluorescence reporter (verified by RT-qPCR) and with pHluorin examine cells at the population and single-cell level. The in vitro assays and Monte Carlo simulations aim at deciphering the underlying molecular mechanism of the observed QLC perturbations.. Both data and methods are carefully presented, with detailed descriptions of the methodological approach (e.g. snf5∆ slow growth phenotype loss and how this was solved). The authors further provide analysis code and resources online, which supports reproducibility. For most experiments, quantification is provided, and statistical tests used appropriately.

We thank the reviewer.

Major comments:1) The one experiment that needs additional explanation/analysis is the in vitro nucleosome remodeling assay. First, do the graphs show single traces or are they averages?

These are averages of 2-4 independent kinetic traces. This is stated in the methods, but we will have also added this information to the legend. There is some day to day variability but the patterns that we highlight are consistent. We will have clarified the revised version.

Similarly, in the supplemental data, the ATPase data were fit, but the fits (and accompanying errors) are not provided-only a representative trace.

This is a representative example. No consistent differences were found. We will clarify in the legend.

Second, the analysis would be more complete if the His mutant of the QLC was included.

We have added these data to figure 5.

Third, he authors could describe their conclusions from these data more clearly. In Figure 5D, the decrease in remodeling with decreasing pH is clear for the wild type traces. It also seems clear that the δ QLC is not pH responsive. However, what is the interpretation of the higher baseline (no TF) signal, including the pattern at the end of the trace, and lower overall remodeling (at all pH). Is it the author’s model that both the QLC and another region in SWI/SNF are required for interaction with VP16, and that the QLC interaction is pH sensitive (thus without the QLC, there is still recruitment but it is not pH sensitive)?

We believe that the small differences in baselines of negative controls is just technical variation.

Recognizing that this assay is not the focus of the paper, and that the authors are careful not to over interpret it, it does provide an important level of mechanistic insight.

We thank the reviewer. Indeed, reconstitution is inherently limited due to its reductionist nature. However, we can distinguish between two key models: modulation of intrinsic chromatin remodeling activity of SWI/SNF versus modulation of recruitment to chromatin. This assay nicely supports the latter mechanism. Furthermore, our new data suggest that the presence of histidines in a low-protonation state is required for interaction with VP16.

2) Since calibration curves are quite central for the pH measurements, exemplary curves should be provided in the supplementary data.

We have added these curves (new Figure 2 supplement 1).

Minor comments:1. While the pADH2 reporter assay is well-controlled with ADH2 mRNA RT-qPCR, it would be interesting to know if the induction effects are also observable on the protein level. Clearly, the authors show that translation of new transcripts is required for pH recovery using cyclohexamide. However, even if no additional experiments (like ADH2 Western blot) are performed, at least some information (possibly from previous publications) about how the findings on the transcriptional level/on the level of ADH2 mRNA induction correspond to the protein level.

We don’t have an antibody for ADH2p, however the main assay we use in the paper requires both transcription and translation of the reporter gene (pADH2-mCherry). The manuscript is very clearly focused on the role of SWI/SNF in transcriptional activation, therefore we won’t investigate possible effects of pH or SWI/SNF on protein stability at this time. This is an interesting topic for future studies though!

2. The statistical test used for RT-qPCR data in Figure 1D should be provided (for example in Figure 4C, the statistical test and the figure are nicely described).

These are Bonferroni-corrected t-tests. We have added this information to the legend.

3. In Figure 4B, it is difficult to see the underlying box plots.

We have made the data points a more transparent to make the box plots more apparent.

4. For the sake of completeness, the authors might want to add how the results differ if culture OD exceeds OD 0.3 before the assays (lines 541+542).

ADH2 induction happens much more quickly if the culture exceeds OD 0.3, and the effects of the SNF5 alleles are less apparent. We believe that this is because the regulatory mechanism under study is important for the very earliest stages of transcriptional remodeling. We have clarified these points in the manuscript.

5. Some typos in the methods section should be revised (e.g. line 710, 755, 758, 759, 766).

We have corrected these typos in the manuscript.

6. It is confusing to refer to a nucleosome as a "model promoter" (as done in the abstract).

We have changed this language to:

“Furthermore, the SNF5 QLC mediated pH-dependent recruitment of SWI/SNF to an acidic transcription factor in a reconstituted nucleosome remodeling assay.”

Reviewer #2 (Significance (Required)):SignificanceThis study represents a conceptual advance. The authors present an exciting model of pH sensing, in which few histidines govern pH-sensitive transcription factor interactions upon carbon starvation in S. cerevisae. By dissecting how a QLC can act as a pH sensor to translate carbon starvation into transcription changes needed for adaptation, the authors provide new insight into disordered protein function (potentially identifying a class of pH sensitive LCDs), and a novel mechanism linking environment (i.e. pH) to transcription control. The authors provide interesting speculation as to how similar mechanisms could be broadly conserved, and highlight examples of pH changes that occur in cells (including human cells). This work is therefore of broad general interest to the field of disordered proteins, transcription, and (potentially) cellular metabolism and its links to disease.

We thank the reviewer.

Expertise lacking: molecular dynamics simulationsReferee Cross-commentingI also agree that the authors should revise this interesting manuscript to address the points raised by the reviewers.

We thank the reviewer.

Reviewer #3 (Evidence, reproducibility and clarity (Required)):The intracellular milieu, including pH, is important for proper functioning of cellular processes. To be able to react to changes, cells need sensing and subsequently correction mechanisms. In the manuscript of Ignacio Gutiérrez et al., the authors showed that in *S. cerevisiae* histidines in a glutamine-rich low complexity sequence (QLC) in the N-terminus of SNF5, a subunit of the SWI/SNF chromatin remodeling complexes, are important for transcription response during carbon starvation. Their model is that a transient acidification leads to protonation of the histidines in the QLC of SNF5 resulting in aconformation change. After a transient acidification, the conformational expansion is reversed leading to interaction with different transcription factors and ultimately to an altered transcriptional response. It is an interesting idea that chromatin remodelers, which are highly important transcription regulators, serve as a pH sensor to adapt gene expression.

Thank you, this indeed is the intended scope of the study.

A bit of the enthusiasm is gone, as the QLC of SNF5 is not conserved in humans, where the authors speculate in the discussion that QLCs of other SWI/SNF subunits could be important here. However, human QLCs are not enriched for histidines according to their analyses.

We have gone back and re-evaluated our bioinformatic analysis with a revised definition of QLCs based on a more rational argument (new Figure 1—figure supplement 2). In this new analysis, QLCs are enriched for histidine in all organisms that we investigated, including humans. This enrichment is especially apparent when comparing QLCs to other low complexity sequences (Figure 1 —figure supplement 3).

As QLC including histidines are frequent across many proteins in several other species, the authors claim that these regions could commonly function as pH sensors, but evidence for this is also missing.

We started to investigate the hypothesis that SNF5 is a pH-sensor, “enrichment for glutamine residues interspersed with histidine residues appears to be conserved sequence feature, both in QLCs in general, and in the N-terminus of SNF5 in particular, implying a possible functional role (40).” It is beyond the scope of our current work to demonstrate pH sensing in multiple organisms, or in a large number of different protein complexes. However, we plan to follow up with these kinds of studies in the future.

While the authors address an important topic and show SNF5 QLC-dependent effects, their data also leave many open questions and the model needs further support of additional data.Major comments:The authors define QLCs as “stretches of low-complexity sequences containing at least 10 glutamines” (p.5). They show an enrichment of alanine, proline and histidine in QLCs in yeast, however, this is not very evident in the SNF5 case directly in the defined QLCs. The N-terminus contains 2 QLCs (26 aa, 64aa), but the relevant histidines are mainly adjacent to the QLCs.

This is a good point. We carefully reconsidered our definition of QLCs in the revised manuscript since our previous working definition was a good starting point, but somewhat arbitrary. Our new definition of a QLC is a sub-region of the proteome in with an average fraction of glutamine residues of 25% or higher (minimum fraction), with a maximum interruption between any two glutamine residues of less than 17 residues, and the where the whole QLC is at least 15 residues in length (minimum length) (new Figure 1—figure supplement 2A). Systematic variation of the maximum interruption size revealed that 17 residues was the value that maximized the number of QLCs and the number of residues found within QLCs, offering an optimally permissive value under the 0.25 or greater fraction of glutamine threshold. Using this new definition of QLCs, we found that the N-terminus of SNF5 is mostly QLC, including all of the relevant histidines in this study (new figure 1A).

Due to additional glutamines in the vicinity, the authors refer here as SNF5 QLC of a 282 amino acid region. This region is further poorly conserved, but a range of ascomycetes showed also high glutamine and several histidine residues in the N-terminus of SNF5. However, it is not clear how evolutionary conserved this feature is across other species.

We have zoomed out to encompass proteomes that more broadly cover the Eukaryotic tree of life. This allowed us to determine that this sequence feature was gained in basidiomycete and ascomycete fungi, indicating conservation for ~1 billion years, but was not present in. the ancestor of the metazoa (animals).

Moreover, the QLC is not conserved in humans and QLCs in humans are not enriched for histidines. It is unclear why the authors only focused on ascomycetes to look at evolutionary conservation. They should extend their studies to provide an idea in which species their proposed model might play a role and what the evolutionary advantage might be to lose the SNF5 QLC. As histidines are also not enriched in human QLCs, the authors should comment on the relevance of their proposed mechanisms for humans (e.g. number of QLCs, QLCs including histidines).

See above. Metazoa don’t appear to have lost the sequence, rather it is gained in the fungi. We also used recent structures to compare BAF and SWI/SNF. Both complexes contain large clusters of low complexity sequences poised near the DNA exiting the nucleosome (see new Figure 5—figure supplement 1). Interestingly, BAF does have one QLC in that position.

While the growth of *S. cerevisiae* under poor carbon sources seems not to depend on SNF5 QLC or the histidines in this region, the authors observe a strongly reduced growth rate if the switch to poor carbon sources is preceded by a 24h glucose starvation. How is the growth rate if they are provided with glucose again after the starvation or pH is kept low for longer? Glucose starvation will also lead to a drop of intracellular ATP concentration and consequently impair ATP-dependent chromatin remodeling. After starvation, the cells may depend more on remodelers to restore the chromatin. The authors should do additional experiments to be able to discriminate the source of reduced cell growth after starvation.

We have added the experiment switching from glucose to starvation and back to glucose Figure 1—figure supplement 6.

With respect to: "do additional experiments to be able to discriminate the source of reduced cell growth after starvation." We believe that it’s likely that failure to induce a key set of starvation response genes to metabolically adapt to carbon starvation is part of the reason for reduced growth (figure 4).

While the C-terminus of SNF5 is interacting with the acidic patch of the nucleosomes (e.g. Han et al., 2020), the N-terminus of SNF5 has been shown to be important for interaction with acidic transcription activators (Neely et al., 2002). Having the snf5Δ mutant, the authors should include also this clone as control in Figure 1—figure supplement 5 and quantify the results, in order to clearly show their statement that deletion of SNF5 QLC is distinct from total loss of the SNF5 gene.

It's very clear from growth phenotypes (e.g. Figure 1—figure supplement 6A) that deletion of SNF5 QLC is distinct from total loss of the SNF5 gene. It has been previously published that complete deletion of SNF5 leads to disruption of the SWI/SNF complex. We believe that repeating these published results in Figure 1—figure supplement 5 would not add much, and because the snf5∆ complex falls apart, the experiment would not be a valid comparison.

Moreover, snf5Δ mutant showed a different phenotypes compared to the N-terminal SNF5 mutants, but authors failed mostly to follow up this mutant in the experiments following figure 1. They should include as much data as possible also for the SNF5 knockout strain, in order to elucidate where the differences arise.

There is a major problem with the snf5Δ mutant strain. It is extremely sick and we believe that it rapidly acquires suppressor mutations (or perhaps undergoes poorly defined epigenetic conversions). For this reason, we have not included a great deal of snf5Δ strain data to prevent confusion. We have added a supplemental figure (Figure 2 —figure supplement 4) showing the snf5Δ results equivalent to figure 2.

The authors followed consequently to investigate gene expression changes genome-wide. This is important for their conclusions; however, the design of the experiment is not entirely clear. What was the rational for choosing these conditions?

These mutants and conditions were designed to test the hypothesis that (A) the N-terminus of SNF5 and Histidines within are important for transcriptional reprogramming and (B) a pH change is required.

Moreover, some transcriptional changes are likely hidden in total RNA-seq and would have benefitted from nascent RNA-seq.

Nascent RNA-seq might reveal some changes in stable mRNAs that we miss in our experiments, but capturing a few more genes doesn’t further our model and it is not our goal to discover every gene regulated by SWI/SNF. Therefore, given the high cost and effort required, we will not perform these experiments.

On top, they showed that ADH2 is only up-regulated in a subset of cells. In order to elucidate this bimodal response, single-cell gene expression analyses would have been more appropriate. The latter two points might have also contributed to the rather limited effects. The results further show that most of SWI/SNF target genes are not altered under these conditions and therefore "normal" SWI/SNF function is mostly maintained.

This is beyond the scope of our study. At the point of response to these reviews, scRNA-seq had only been successfully attempted twice in *S. cerevisiae* (PMID: 31985403, PMID: 32420869). Each of these experiments was an *eLife* paper. We don’t believe that these experiments would further our model. Therefore, given the high cost and effort required, we will not perform these experiments.

In order to investigate the mechanistic effects of the loss or mutation of the SNF5 QLC, the authors should extend their in vitro assays and investigate if it is leading to altered protein-protein interaction (as proposed), altered chromatin binding (e.g. by ChIP-seq) or affects the SWI/SNF remodeling activity/ chromatin accessibility genome-wide (e.g. by ATAC-seq or MNase-seq) in yeast.

Again, our manuscript is not about the genomics of SWI/SNF, and this topic has been extensively researched before, including by coauthors of this study. We already know that recruitment of SWI/SNF to chromatin is the fundamental molecular mechanism for transcriptional control. None of the above experiments would significantly further the model proposed in the title and abstract. Therefore, given the high cost and effort required, we will not perform these experiments. Nevertheless, we have undertaken more focused ChIP experiments to test the recruitment of SWI/SNF to the ADH2 promoter (Figure 1—figure supplement 8)

Also in their model, they assume that the histidines may partially be still protonated after pH recovery leading to altered interaction partners. However, experimental evidence is missing for that and what the impact on this low complexity region and the interactions with other proteins are.

It’s unclear what the experiment would be here. It's impossible to know the in vivo protonation state of SNF5, but we believe that it’s reasonable to propose that this protonation state changes over the intracellular pH ranges measured in our experiments, given our knowledge of chemistry and the pKa of the histidine side chain.

The authors need to add statistical information, e.g. number of replicates, type of replicates, error bar information, statistical tests used.

We have added this information.

Minor comments:The authors should describe in the method section how they assessed growth rate.

We have added these methods.

The authors have several formatting or reference inconsistencies (e.g. Figure 1 sup 2C (p.6));

We have addressed this formatting.

Figure 2A, C instead of Figure 2A, B (p.9);

This is actually correct – 2B is a guide to help interpret the quantification in 2C.

Figure 3—figure supplement 3 and 4 (p.11);

This change is included in the current version of the manuscript.

Figure 3B, right (p.11);

This change is included in the current version of the manuscript.

Figure 5 (within figures Ph 7.6, in text Ph 7.5),

This change is included in the current version of the manuscript.

Yudkovsky et al. 1999 (p.17), Supplementary Tables) and should improve their labeling for certain graphs (e.g. Figure 1D, Figure 2A (addition of % values per quadrant), Figure 2C.

We have added these values to the revised manuscript.

Figure 2C, 3B: pHi – legend for black and grey lines are missing. Are these two replicates? If so, the authors should also show the third replicate mentioned in the legend (2C).

We have clarified the legend.

Data for the experiment with sorbic acid are not shown (p.11).

We are not sure which data the reviewer is referring to here.

Please indicate which clustering was used for Figure 4E and which “manual curation” was performed.

We used Euclidean clustering. Manual curation was when to removed small clusters that didn't have significant GO hits, and to consolidate clusters that had similar behavior. We have described these methods in more detail.

Figure 5B: There seem to be slight differences between WT and the mutant. An overlay of them would be nice to better compare the effects.

These slight differences are technical variation. Overlay of WT and dQ at pH 7.6 is shown in Author response image 1.

**Author response image 1. sa2fig1:** 

Figure 5—figure supplement 1: A, How was significance determined?

In fact, it is difficult to speak to significance here. We will rephrase to say “only minor changes in ATPase activity were observed”. This is consistent with no impact of pH on nucleosome sliding activity in the absence of activator recruitment.

Reviewer #1 (Significance (Required)):Cells need to sense and adjust changes in the intracellular environment. The authors propose that QLCs in important gene expression regulators – here a subunit of the SWI/SNF chromatin remodeler – can sense alterations in pH and consequently adjust gene expression. While the concept is interesting, further experiments are required to prove their model. Also due to a lack of conservation of this region, this particular QLC is not relevant for human. It would be great, if the authors could address the impact of QLCs in general for pH sensing and response and investigate, if similar mechanisms hold true for humans.

We agree that extension of the project would be very interesting. However, the title of the paper clearly indicates that the scope is for SWI/SNF. We are analyzing further pH sensing mechanisms in current/future work. We think that this demonstration that a crucial chromatin remodeling complex is a pH sensor is very interesting.

Referee Cross-commentingI agree with reviewer 1 and 2 to provide the authors the opportunity to revise their manuscript and address the points raised.